# Serotonin signaling modulates growth and motility in juvenile *Fasciola hepatica*

Emily Robb [1]*, Sarah Muise[1], Lana Watt[1], Rebecca Armstrong[1], Duncan Wells[1],
Paul McCusker[1], John Harrington[2], Andreas Krasky[3], Paul M. Selzer[4], Nikki J. Marks[1],
Aaron G. Maule[1]*

**1** School of Biological Sciences, Queen's University Belfast, Belfast, United Kingdom of Great Britain and Northern Ireland, **2** Molecular Parasitology, Boehringer Ingelheim Animal Health, Athens, Georgia, United States of America, **3** Research and Development, Boehringer Ingelheim Animal Health, Ingelheim am Rhein, Germany, **4** Parasitology, Boehringer Ingelheim Animal Health, Ingelheim am Rhein, Germany

* e.robb@qub.ac.uk (ER); a.maule@qub.ac.uk (AGM)

## Abstract

*Fasciola hepatica* causes fasciolosis, a parasitic disease that poses significant animal and human health challenges. Control relies on flukicides, most of which are adulticides, with only triclabendazole effective against the pathogenic migratory juvenile. Classical neurotransmitter pathways are widely targeted by anthelmintics yet remain underexplored for flukicide development. Here we explore the importance of serotonin (5-HT) signaling in juvenile fluke. *In silico* analyses confirmed all *F. hepatica* life stages express a complete 5-HT signaling pathway encompassing genes encoding proteins for 5-HT synthesis, transport, and reuptake, as well as five putative 5-HT G protein-coupled receptors (GPCRs). Homology and binding motif analyses supported the presence of two 5-HT$_1$ (Fh5HT$_{1A}$, Fh5HT$_{1B}$) and three 5-HT$_7$ (Fh5HT$_{7A}$, $_{-7B}$, $_{-7C}$) GPCRs. Immunocytochemistry and *in situ* hybridization revealed widespread neuronal expression of 5-HT, its synthetic enzyme tryptophan hydroxylase (FhTPH), and the GPCR Fh5HT$_{7C}$. 5-HT addition stimulated juvenile fluke motility; consistent with this observation, serotonin reuptake inhibition, which causes 5-HT persistence at synaptic junctions, also enhanced juvenile movement. Silencing of FhTPH, a key enzyme in 5-HT synthesis, blunted juvenile motility, a phenotype reversed by the addition of 5-HT. Silencing the fluke vesicular monoamine transporter (FhVMAT), which packages 5-HT into synaptic vesicles, reduced juvenile motility, whilst silencing the 5-HT reuptake transporter (FhSERT) which recycles synaptic 5-HT increased juvenile motility and growth, consistent with 5-HT accumulation enhancing effects. Whilst combinatorial silencing of Fh5HT$_1$ receptors reduced fluke motility, silencing Fh5HT$_7$ receptors led to a greater reduction in motility. Exogenous addition of 5-HT partially rescued motility deficits of juveniles with silenced Fh5HT$_1$ receptors, but 5-HT excitation was abolished in Fh5HT$_7$-RNAi juveniles, exposing their importance to fluke motility. Notably, sustained 5-HT exposure promoted juvenile growth, but these effects were not blunted by receptor-RNAi. The findings emphasize a central

**Data availability statement:** All data to replicate the findings (including raw data for figures) are provided in the manuscript and Supporting information.

**Funding:** Funding for this study was provided by Boehringer Ingelheim (to AGM) the Biotechnology and Biological Sciences Research Council (BB/T002727/1 to AGM and NJM). The funders had no role in study design, data collection and analysis, decision to publish, or preparation of the manuscript.

**Competing interests:** The authors have declared that no competing interests exist.

role of serotonin signaling in both juvenile motility and growth, exposing novel aspects of receptor function and encouraging therapeutic exploitation for liver fluke control.

## Author summary

The liver fluke, *Fasciola hepatica*, causes fasciolosis, a neglected tropical disease that poses a significant burden on human and animal health. There is no vaccine for fasciolosis and treatment relies on a single drug, triclabendazole, to control the early stages of infection which cause liver pathology whilst migrating through the mammalian host. Single drug reliance has increased the incidence of drug resistance in both human and animal populations, such that there is a pressing need for the characterization of novel drug targets and development of new anthelmintics targeting liver fluke. The focus of this research is to examine the role of the serotonin signaling system of liver fluke, bridging a gap in knowledge to enable the exploitation of this signaling pathway for flatworm drug development. Here, bioinformatic analysis has characterized the pathway components and receptors in multiple clinically relevant flatworm parasite species. Chemical and functional genomic methods have been used to prove the integral function of serotonin in liver fluke biology, regulating motility and growth, both essential for parasite infection and survival. This work provides data that help validate the serotonergic system of liver fluke as a potential target for future anthelmintic development.

## Introduction

Fasciolosis, caused by the liver fluke *Fasciola hepatica*, is a global issue affecting both livestock and humans, with agricultural losses estimated at $3.2 billion annually [1]. As a neglected tropical disease, it impacts approximately 17 million people in some of the world's poorest regions. The most effective treatment for fasciolosis is the anthelmintic drug triclabendazole [2]. Triclabendazole is the only drug effective against the early juvenile stages of the parasite, which cause the most severe damage during their migration through the hosts liver [3]. However, the heavy reliance on triclabendazole for treatment has led to increasing cases of drug resistance in both human and livestock populations, underscoring the need for the development of new drugs to combat these parasites [4,5].

The flatworm nervous system has well developed classical neurotransmitter and neuropeptidergic systems, which are integral to regulating biological activities essential for parasite survival, including motility, reproduction, growth, and development [6]. Research has identified these systems as a source of potential druggable targets. In particular, the neuropeptidergic system has attracted considerable attention, with numerous studies reporting drug target validation from within this pathway [6,7].

Notably, neurotransmitter pathways have been successfully targeted in nematode parasites, with highly effective anthelmintic drugs such as levamisole and ivermectin [8]. However, the relative sparsity of studies characterizing analogous flatworm neurotransmitter pathways has hindered similar progress in drug development for flatworm parasites [9]

Serotonin (5-HT), an ancient signaling molecule conserved across metazoans, is one of the most extensively studied neurotransmitters, particularly in higher animals and humans. Its roles extend beyond modulating neuronal activity to include the regulation of processes such as metabolism and immune responses [10,11]. In vertebrates, endogenous 5-HT synthesis occurs through a well-characterized pathway involving the enzymes tryptophan hydroxylase (TPH) and aromatic L-amino acid decarboxylase (AADC), which convert tryptophan into 5-HT [12,13]. Once synthesized, 5-HT is packaged into synaptic vesicles by vesicular monoamine transporters (VMAT) and released into the synaptic cleft [12,13]. 5-HT signals via a 5-HT-gated cation-selective ion channel (5-HT3) and/or G protein coupled receptors (GPCRs; 5HTRs) which can couple to multiple G protein families (e.g., $G_s$, $G_{i/o}$, $G_{q/11}$, and $G_{12/13}$). Depending on receptor subtype, this signaling can result in increases or decreases in cyclic adenosine monophosphate (cAMP) production by adenylate cyclase, as well as activation of additional downstream pathways such as phospholipase C and Rho-mediated signaling [12,13]. 5-HT signaling is terminated by 5-HT reuptake transporters (SERT), which clear the neurotransmitter from the synapse [12,13]. In parasitic helminths, bioinformatic studies suggest that a 5-HT synthesis and signaling pathway similar to that of vertebrates is conserved. However, while GPCR-mediated 5-HT signaling has been characterized in helminths, a 5-HT-gated ion channel specific to flatworm parasites has yet to be identified [14,15]. Parasitic flatworms have also been shown to actively transport exogenous 5-HT from hosts across their tegument further suggesting functional importance for parasite survival within a host [16–18].

Despite an extensive history of study, the function of 5-HT in flatworm biology remains poorly characterized for drug target exploitation. Early studies show high expression of 5-HT in both the central and peripheral nervous system with prevalence in neurons linked with innervation of the digestive tract and body wall muscles of adult *F. hepatica* [19,20]. Rich serotonin immunoreactivity was also associated with the oral and ventral suckers and surrounding nerve fibres of *F. hepatica* [20]. Evidence strongly supports serotonin's role in modulating neuromuscular function and motility in flatworms. For instance, 5-HT excites muscle strips in liver fluke [21,22], as well as in other parasitic flatworms such as the cestode *Hymenolepis diminuta* [23] and monogenean *Diclidophora merlangi* [24]. Whole-worm motility assays further underscore 5-HT's role in modulating flatworm motility, as it stimulates larval motility in the blood fluke *Schistosoma mansoni* [18] and other flatworm species [23,24]. Direct actions on muscle were supported by the fact that 5-HT was found to be critical for dispersed muscle fiber contraction in *S. mansoni* [25]. Molecular studies have further corroborated serotonin's importance as a regulator of parasite movement. For example, RNA interference (RNAi)-based silencing of the 5-HT reuptake transporter (smSERT) in *S. mansoni* led to a marked increase in schistosomula motility, while knockdown of a serotonin receptor (sm5HTR) resulted in a significant decrease in motility [18,26]. The role of 5-HT in modulating flatworm motility encourages consideration as a therapeutic target in parasitic flatworms.

Beyond its role in neuromuscular control, 5-HT has been implicated in metabolic activity and development in parasitic flatworms. In several species, serotonin enhances carbohydrate metabolism and glucose uptake, increasing ATP availability to support energy-intensive processes such as muscle contraction and growth [27–29]. Emerging evidence further suggests that 5-HT contributes to cell proliferation and developmental progression in flatworms [30–32]. For example, 5-HT has been shown to stimulate metacestode development and cell proliferation in *Echinococcus multilocularis,* highlighting a potential role in cestode growth and survival [33]. Together, these observations suggest that serotonin signaling may influence not only motility but also growth-related processes in parasitic flatworms, supporting its evaluation as a multifunctional therapeutic target.

There is substantial evidence to support the hypothesis that 5-HT signaling is critical to a variety of essential systems in flatworms. Coupled with pharmacological studies indicating the druggability of parasite-specific 5-HT receptors [34], this signaling system presents a promising avenue for the identification of novel therapeutic targets. However, critical

knowledge gaps persist, as the role and importance of multiple components of the 5-HT signaling systems in flatworms remain poorly characterized, impeding drug development efforts.

This study aims to characterize the 5-HT signaling pathway for the liver fluke, *F. hepatica*. Functional genomics complemented by expression analyses and the chemical validation of 5-HT pathway components underscore their critical role in regulating the motility of the juvenile liver fluke, whilst simultaneously exposing a broader role for serotonin in parasite growth. Additionally, bioinformatic analyses identified 5-HT-gated G protein-coupled receptors (GPCRs) across multiple clinically relevant flatworm species, encouraging exploitation of this pathway for the development of novel flukicides/anthelmintics.

## Results and discussion

### *F. hepatica* juveniles express a complete gene set for 5-HT signaling

*In silico* analysis (Fig 1A) revealed significant conservation of 10 genes linked to 5-HT signaling in *F. hepatica*. These include five genes encoding proteins involved in 5-HT synthesis, transport, and reuptake: a tryptophan hydroxylase (FhTPH); two aromatic L-amino acid decarboxylases (FhAADC); a vesicular monoamine transporter (FhVMAT); and a 5-HT reuptake transporter (FhSERT) (S1 Table). Additionally, five putative serotonin G protein-coupled receptor (GPCR) genes (Fh5HTR) were identified (Fig 1B and 1C and S1 Table). All identified genes exhibit high conservation of functional domains, as observed in orthologues from humans [35–38] and blood fluke, *Schistosoma mansoni* (Fig 1D) [15,18,26,39]. Notably, *F. hepatica* expresses two distinct AADC enzymes (FhAADC1 & FhAADC2), whereas humans possess only one. This gene duplication, which has also been observed in other flatworm species, may reflect an additional role in dopamine signaling [15]. These findings align with genomic studies demonstrating conservation of 5-HT signaling components across diverse flatworm species [15,40]. Transcriptome analysis revealed peak abundance of 5-HT signaling pathway genes during the newly excysted juvenile (NEJ) stage, suggesting a role for 5-HT in the highly motile infective and migratory phases of the parasite's life cycle (Fig 1E and S1 Table), consistent with the hypothesis that 5-HT is essential for neuromuscular function. Importantly, *in vitro* cultured juveniles represent a transcriptionally distinct developmental state relative to both newly excysted juveniles and host-derived juveniles, as demonstrated previously [41]. Interestingly, the synthesis enzymes FhTPH and FhAADC also show elevated abundance during the egg stage, supporting a non-neuronal role for 5-HT in egg development and biology (Fig 1E and S1 Table). This observation suggests an important role for 5-HT in the egg stage, where the assimilation of exogenous 5-HT is likely limited, necessitating endogenous production to fulfil developmental requirements. Five putative 5-HT-GPCRs were identified in *F. hepatica;* FhHiC23_g2340, FhHiC23_g7880, FhHiC23_g4917, FhHiC23_g1196 and FhHiC23_g115 (Fig 1B and S1 Table). Consistent with the literature, a 5-HT-gated ion channel was not identified as part of this analysis [15]. Alignments with characterized human 5HT$_1$ (Q5ZGX3) and functionally characterized *S. mansoni* 5HT$_7$ (Smp_126730; [26]) 5-HT-gated GPCRs show conservation of key motifs essential for ligand binding and receptor activation, including the DVXXCT motif in transmembrane (TM) 3, the WXXF motif in TM6, and the canonical DRY motif at the cytoplasmic end of TM3 (Ballesteros–Weinstein (BW) positions 3.49–3.51) [42,43] (Fig 2A).

### 5-HT-gated GPCRs are conserved across clinically relevant flatworms

A further 75 flatworm 5-HT-gated GPCRs were identified across clinically significant cestode and trematode species (S2 Table), representing the most extensive dataset of flatworm 5-HT receptors to date. All 75 identified flatworm GPCRs exhibit conservation of 5-HT-associated ligand binding and receptor activation motifs, as previously characterized for *F. hepatica* (S1 Fig). Notably, a greater number of 5-HT receptors were found in trematode species (average = 4) compared to cestode species (average = 2) (Fig 2B). Interestingly, Wheeler et al. [47] reported that the expansion of chemoreceptor families in nematodes correlates with the presence of environmental and extra-host stages. While 5-HT GPCRs are not

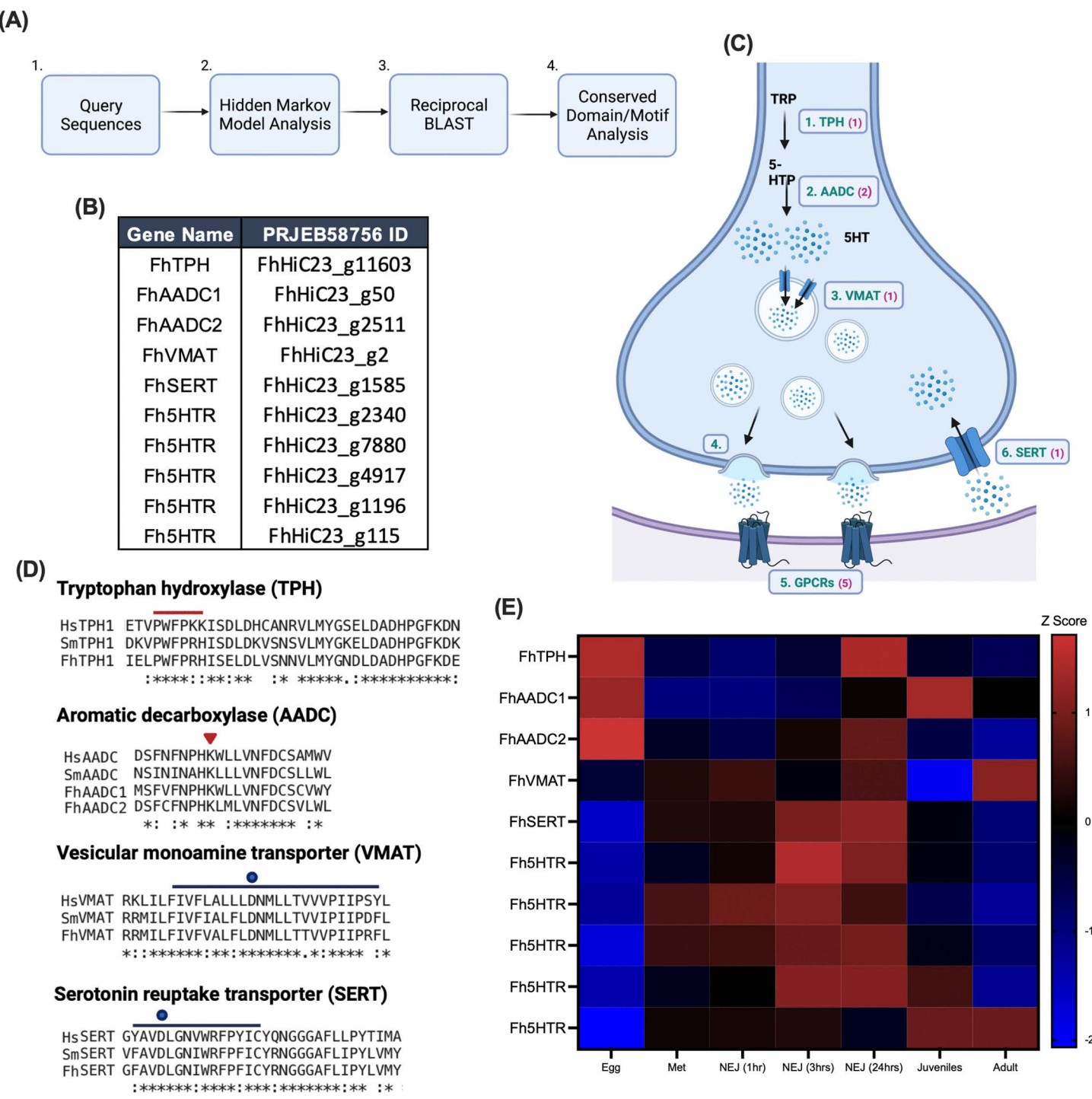

**Fig 1.** *Fasciola hepatica* **possesses serotonin signaling pathway genes. (A)** Schematic representation of the bioinformatic pipeline used to identify serotonin signaling genes in *F. hepatica*. **(B)** Gene identifiers (IDs) of serotonin-related components identified in the *F. hepatica* genome assembly PRJEB58756. Abbreviations: FhTPH = tryptophan hydroxylase; FhAADC = aromatic L-amino acid decarboxylase; FhVMAT = vesicular monoamine transporter; FhSERT = serotonin reuptake transporter; Fh5HTR = serotonin-gated G protein-coupled receptors (GPCRs). **(C)** Illustration of serotonin signaling at a synapse. (1) The precursor amino acid tryptophan (TRP) is converted to 5-hydroxytryptophan (5-HTP) by tryptophan hydroxylase (TPH). (2) 5-HTP is further converted to serotonin (5-hydroxytryptamine, 5-HT) by aromatic L-amino acid decarboxylase (AADC). (3) 5-HT is sequestered into synaptic vesicles via vesicular monoamine transporters (VMAT). (4) Vesicles fuse with the synaptic membrane, releasing 5-HT into the synaptic cleft.

(5) Released 5-HT binds to GPCRs on the postsynaptic neuron to propagate signaling. Note: although serotonin can also interact with ligand-gated ion channels (LGICs), no 5-HT-sensitive LGICs were identified in the *F. hepatica* genome. Magenta-bracketed numbers indicate the number of encoding genes identified in genome PRJEB58756. *Created in BioRender. McCusker, P.* (2026) https://BioRender.com/28zes5h. **(D)** Sequence alignments of *F. hepatica* serotonin signaling genes with homologous sequences from *Homo sapiens* (Hs-) and *Schistosoma mansoni* (Sm-). Red lines highlight conserved catalytic domains in TPH, and red arrowhead indicates pyridoxal-phosphate binding site in AADC. Blue lines represent transmembrane domain 1 of VMAT and SERT, while a blue dot marks a key aspartic acid residue implicated in ligand binding. **(E)** Heatmap displaying expression levels of *F. hepatica* serotonin signaling genes across different life cycle stages (egg to adult). Z-scores were calculated from transcripts per million (TPM) data obtained from WormBase ParaSite (release 54) and generated by Cwiklinski et al. [44].

typically considered chemosensory receptors, it is possible that variation in the number of 5-HT receptors reflects differences in life history traits, tissue complexity, or neuromodulatory demands between parasite groups, however, any link between receptor number and lifestyle remains speculative and requires further functional investigation. Recently, Camicia et al. [48] presented evidence suggesting that cestode $5HT_1$ receptors may have evolved distinct motif patterns, potentially making them harder to detect using bioinformatic approaches. As a result, it is possible that some sequences may have been excluded from this analysis.

### *F. hepatica* express $5HT_1$ and $5HT_7$ type GPCRs

Understanding of 5-HT receptors in parasitic flatworms has been informed by a combination of functional and bioinformatic studies, with most detailed functional characterization coming from work in schistosomes [49–51]. Several 5-HT GPCRs have been examined using heterologous expression and pharmacological approaches in Schistosoma, providing insight into receptor signaling properties and ligand specificity [49–51]. More recently, heterologous expression and functional characterization of individual 5-HT receptors have also been reported in cestodes [48,52]. In contrast, for many other parasitic flatworms, functional data for serotonergic GPCRs remain limited, and receptor identification and classification have relied primarily on genomic and *in silico* analyses [15,40,52–54]. As a result, receptor classification has largely been conducted through homology-based approaches, comparing flatworm receptors to their vertebrate counterparts [45]. Through such methods, most 5-HT receptors identified in parasitic flatworms have been classified into the $5\text{-HT}_1$ and $5\text{-HT}_7$ clades ([15,26,48,55]. Functionally, $5HT_1$ receptors couple to $G_{i/o}$ proteins to inhibit adenylate cyclase and decrease cAMP, whilst activation of $5HT_7$ receptors coupled to $G_s$ proteins activate adenylate cyclase and increase intracellular cAMP [48].

Phylogenetic analysis grouped two *F. hepatica* 5-HT-GPCRs, FhHiC23_g2340 and FhHiC23_g7880, alongside human and *S. mansoni* $5HT_1$ receptors, previously identified via *in silico* analysis (Fig 2C) [15]. Three further *F. hepatica* 5-HT-activated GPCRs, FhHiC23_g4917, FhHiC23_g1196 and FhHiC23_g115, were phylogenetically associated with the human $5\text{-HT}_7$ receptor (P34969) and functionally characterized $5\text{-HT}_7$ receptor (Smp_126730) from *S. mansoni* [26] suggesting these are $5\text{-HT}_7$ family serotonin-gated GPCRs (Fig 2C). Further corroboration using CLANs and clustering analysis revealed two distinct groups of flatworm 5-HT-gated GPCRs (Fig 2D and S2 Table). Cluster 1 contains the three *F. hepatica* GPCRs tentatively assigned to the $5\text{-HT}_7$ receptor family (FhHiC23_g4917, FhHiC23_g1196, and FhHiC23_g115). This cluster shows close sequence similarity with *S. mansoni's* functionally characterized $5\text{-HT}_7$ receptor (Smp_126730), but notably, the human $5\text{-HT}_7$ receptor did not cluster with any flatworm GPCRs (Fig 2D). This sequence divergence between human and flatworm receptors could highlight an opportunity to selectively target flatworm-specific 5-HT receptors with anti-parasitic drugs. Cluster 2 includes *F. hepatica's* FhHiC23_g2340, identified as a likely $5HT_1$-type receptor, and shows strong clustering with *S. mansoni* $5\text{-HT}_1$ receptor genes (Smp149770 and Smp197700). Interestingly, while FhHiC23_g7880 groups more closely with $5HT_1$ receptors, it did not cluster distinctly with any characterized 5-HT receptor family in the CLANs analysis (Fig 2D). Motif analysis confirmed that all *F. hepatica* 5-HT receptors share a conserved pattern of cysteine (C) and threonine (T) residues in transmembrane 3, indicative of either the $5\text{-HT}_1$ or $5\text{-HT}_7$ receptor clades (S2 Fig) [48]. Notably, FhHiC23_g2340 and FhHiC23_g7880 exhibit a conserved tryptophan (W) within

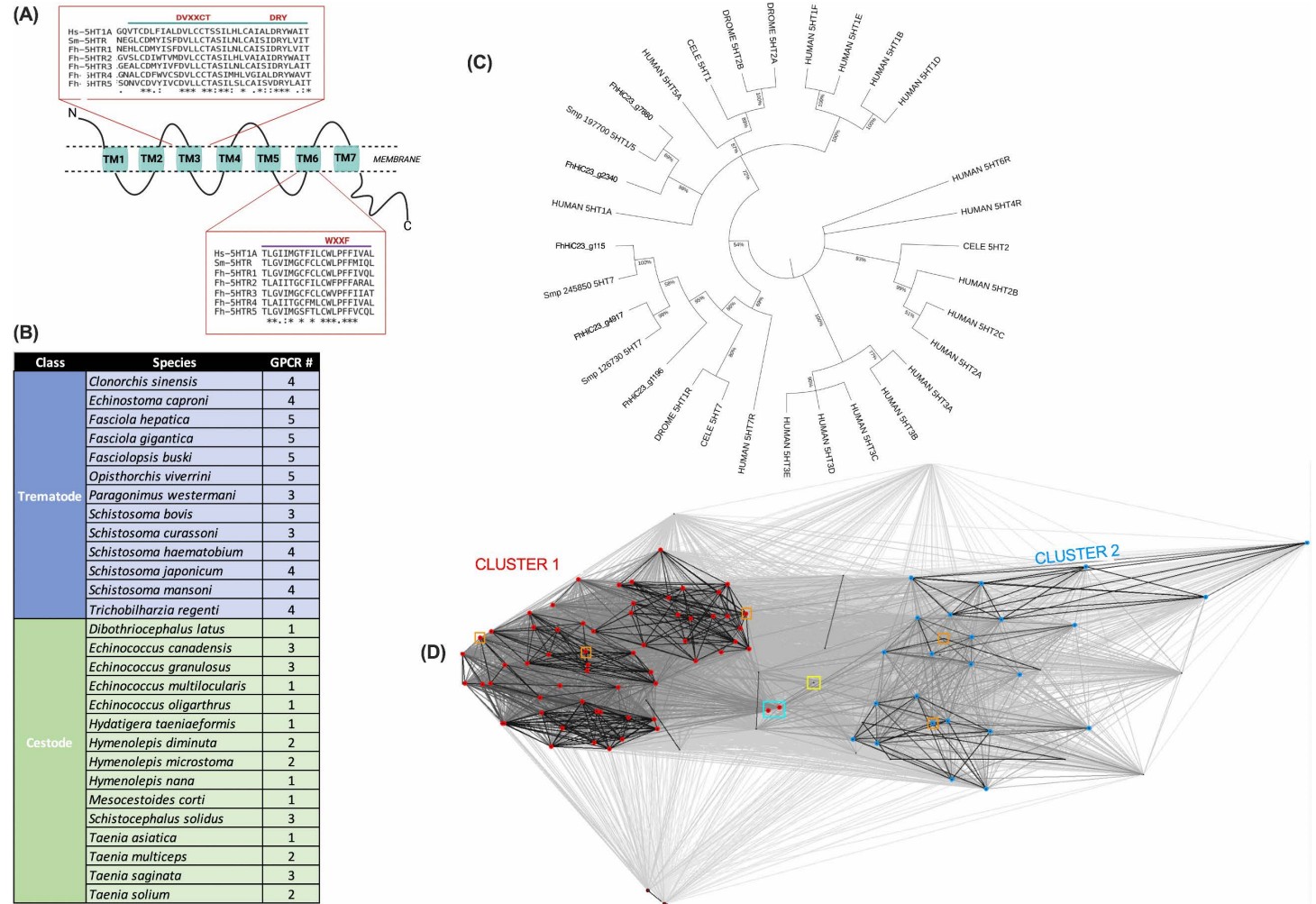

**Fig 2. Comparative and phylogenetic analysis of serotonin-gated G protein-coupled receptors (GPCRs) in *Fasciola hepatica* and other parasitic flatworms.** (A) Multiple sequence alignment of *F. hepatica* serotonin GPCRs with homologous sequences from *Homo sapiens* (Hs-) and *Schistosoma mansoni* (Sm-). The green line highlights transmembrane domain three (TM3), while the purple line highlights transmembrane domain six (TM6). Conserved serotonin-binding motifs (DVXXCT and WXXF) essential for ligand recognition, as well as the canonical 'DRY' motif critical for GPCR activation, are outlined in red. (B) Summary table displaying the number of serotonin-gated GPCRs identified across clinically relevant trematode and cestode species. Genomic data were obtained from WormBase ParaSite (version 19) and analyzed for the presence of serotonin GPCR homologs. The species investigated include *Echinococcus canadensis* (PRJEB8992), *Echinococcus granulosus* (PRJNA182977 - G1), *Echinococcus multilocularis* (PRJEB122 - Java), *Echinococcus oligarthrus* (PRJEB31222 - DaMi1*), Hymenolepis diminuta* (PRJEB507 - Denmark), *Hymenolepis microstoma* (PRJEB124), *Hymenolepis nana* (PRJEB508 - Japan), *Mesocestoides corti* (PRJEB510 - Specht & Voge 1965), *Taenia asiatica* (PRJEB532 - South Korea), *Taenia multiceps* (PRJNA307624 - Gns01), *Taenia saginata* (PRJNA71493 - TSAYD01), *Taenia solium* (PRJNA170813 - Mexico), *Dibothriocephalus latus* (PRJEB1206*), Spirometra erinaceieuropaei* (PRJEB1202), *Schistocephalus solidus* (PRJEB527 - NST_G2), *Hydatigera taeniaeformis* (PRJEB534 - Spain/Canary Islands), *Clonorchis sinensis* (PRJNA386618 - Cs-k2), *Schistosoma bovis* (PRJNA451066), *Schistosoma japonicum* (PRJNA520774 - HuSjv2), *Fasciolopsis buski* (PRJNA284521 - HT), *Fasciola gigantica* (PRJNA230515 - Uganda_cow_1), *Schistosoma haematobium* (PRJNA78265), *Schistosoma curassoni* (PRJEB519 - Senegal/Dakar), *Schistosoma mansoni* (PRJEA36577*), Trichobilharzia regeni* (PRJEB44434 - tdTriRege1.1), *Opisthorchis viverrini* (PRJNA222628), *Paragonimus westermani* (PRJNA454344), and *Echinostoma caproni* (PRJEB1207 - Egypt). (C) Maximum likelihood phylogenetic tree depicting the evolutionary relationships between *F. hepatica* serotonin-gated GPCRs and homologous GPCR sequences from other species. Reference sequences were obtained from *Homo sapiens* (5-HT1A-F, 5-HT2A-C, 5-HT3A-E, 5-HT5A, and 5-HT7), *Caenorhabditis elegans* (5-HT1, 5-HT2, and 5-HT7), and *Drosophila melanogaster* (5-HT1 and 5-HT2A/B) through UniProt [45]. Additionally, published *S. mansoni* GPCR sequences (Smp_197700, Smp_245850, and Smp_126730) were included based on prior literature [references 14,29,46]. (D) CLANS (Cluster Analysis of Sequences) diagram visualizing the clustering relationships of serotonin-gated GPCRs from all examined parasitic flatworm species. Clustering groups are color-coded based on sequence similarity and evolutionary relationships. Red shows flatworm 5-HT GPCR cluster 1 sequences; blue shows flatworm 5-HT GPCR cluster 2 sequences; orange boxes show clustering of *F. hepatica* 5-HT GPCR sequences; yellow box shows human 5-HT$_1$ GPCR sequence; cyan box shows *D. melanogaster* 5-HT$_2$ GPCR sequences.

transmembrane domain 3 (BW 3.28 [43]), a defining feature of $5HT_1$ receptors, whereas the remaining receptors possess tyrosine (Y) at this position, consistent with $5HT_7$ receptors (S2 Fig). Furthermore, within transmembrane helix 5, residues corresponding to BW positions 5.42 and 5.46 [43] are serine (S) and alanine (A) residues, in FhHiC23_g2340 and FhHiC23_g7880, consistent with 5-HT$_1$ receptor classification (S2 Fig). In contrast, the other three receptors have alanine (A) residues at both positions, matching the 5-HT$_7$ receptor profile as described by Camicia et al. [48] (S2 Fig).

The binding of 5-HT to GPCR receptors induces a conformational change, triggering the release of G proteins to propagate a signaling cascade [46]. To predict the most likely G protein coupling preferences of *F. hepatica* serotonin receptors, *in silico* analysis was performed based on the presence of known structural motifs associated with G protein selectivity [56] (S3 Fig). FhHiC23_g2340 and FhHiC23_g7880 possessed motifs predictive of preferential coupling to $G_{i/o}$ family G proteins, a hallmark of $5HT_1$ receptors, further supporting their classification as 5-HT$_1$ type (S3 Fig). The three 5-HT-gated GPCRs phylogenetically grouped as 5-HT$_7$ type exhibited motifs with varying levels of predicted G protein coupling specificity (S3 Fig). FhHiC23_g4917 displayed motifs predictive of exclusive coupling to Gs proteins, which are associated with 5-HT$_7$ receptor signaling, supporting classification as a 5-HT$_7$ receptor (S3 Fig). Although FhHiC23_g115 also showed positive coupling to $G_{i/o}$ proteins, it exhibited highest predicted specificity for Gs proteins, further validating its designation as a 5-HT$_7$ receptor (S3 Fig). Interestingly, FhHiC23_g1196 showed motifs predictive of potential coupling to all G proteins analyzed, exposing the potential for promiscuity in its interactions and biological function (S3 Fig).

Based on the analysis outlined, *F. hepatica* 5-HT GPCRs were designated as follows; FhHiC23_g2340- $5HT_{1AFhep}$, FhHiC23_g7880- $5HT_{1BFhep}$, FhHiC23_g4917- $5HT_{7AFhep}$, FhHiC23_g115- $5HT_{7BFhep}$ and FhHiC23_g1196-$5HT_{7CFhep}$.

## Modelling of *Fasciola hepatica* 5-HT-gated G protein coupled receptors

To probe the presence of 5-HT binding sites and determine structural similarity to human homologues, *F. hepatica* 5-HT GPCRs were modelled using PyMOL [57] (Fig 3A-3E). Ligand docking analysis confirmed all receptors could bind 5-HT ligand in a similar region to human 5-HT receptors [58] (Figs 3A-3E and S4). Comparison of predicted binding sites of 5HT$_7$ receptors identified a conserved set of residues hypothesized to be key for 5-HT ligand binding (Fig 3F), including Aspartate (D), Valine (V) and Threonine (T) in transmembrane 3 (BW~3.32–3.34 [43]), Glutamine (Q), Alanine (A) and Threonine (T) in transmembrane 5 (BW 5.42 & 5.46 [43]) and Glutamine (G)/Alanine (A), Phenylalanine (F) in transmembrane 6 (BW 6.51–6.55 [43]) (Fig 3F). Similar analysis of 5HT$_1$ receptors also revealed conservation of key residues within transmembrane helices 3, 5 and 6 (Fig 3F). In TM3, Aspartate (D), Valine (V) and Cysteine (C) (BW~3.32–3.34 [43]) residues are consistent with known amine-binding sites in class A GPCRs [59]. In TM5, Phenylalanine (F) or IsoLeucine (I) and Serine (S) (BW 5.42-5.46 [43]) residues were observed, while TM6 contained Proline (P), Phenylalanine (F), Alanine (A) and Leucine (L) (BW~6.48–6.55 [43]), implicated in ligand stabilization and receptor activation [59] (Fig 3F). Together, these conserved features support a shared structural basis for 5-HT recognition between liver fluke and vertebrate 5-HT receptors. Alignment analysis compared the structure of *F. hepatica* 5-HT GPCRs to human 5-HT GPCRs, generating a root mean square deviation (RMSD) value of similarity - the lower the RMSD value, the higher the structural similarity between two proteins with an RMSD value of <2 Å considered highly similar. *F. hepatica* 5HT$_1$ receptors had RMSD values of 0 ($5HT_{1BFhep}$) and 0.684 ($5HT_{1AFhep}$) which indicates high similarity - indeed an RMSD value of 0 suggests the absence of conformational changes (S4 Fig). *F. hepatica* 5HT$_7$ receptors showed RMSD values of 0.684 ($5HT_{7AFhep}$), 0.760 ($5HT_{7BFhep}$) and 0.791 ($5HT_{7CFhep}$), again showing structural similarity with human 5-HT GPCRs (S4 Fig).

## 5-HT signaling components are highly expressed in nervous system of *F. hepatica*

Immunocytochemistry demonstrates widespread 5-HT expression in the central and peripheral nervous systems of 3-week-old juvenile *F. hepatica* (5-HT expression has been reported previously in the adult life stages [20], supporting its extensive role in neuromuscular and physiological regulation (Fig 4A and 4B). 5-HT immunoreactivity (IR) is prominently localized along the ventral nerve cords and in large neuronal cell bodies in and around the cerebral ganglia and

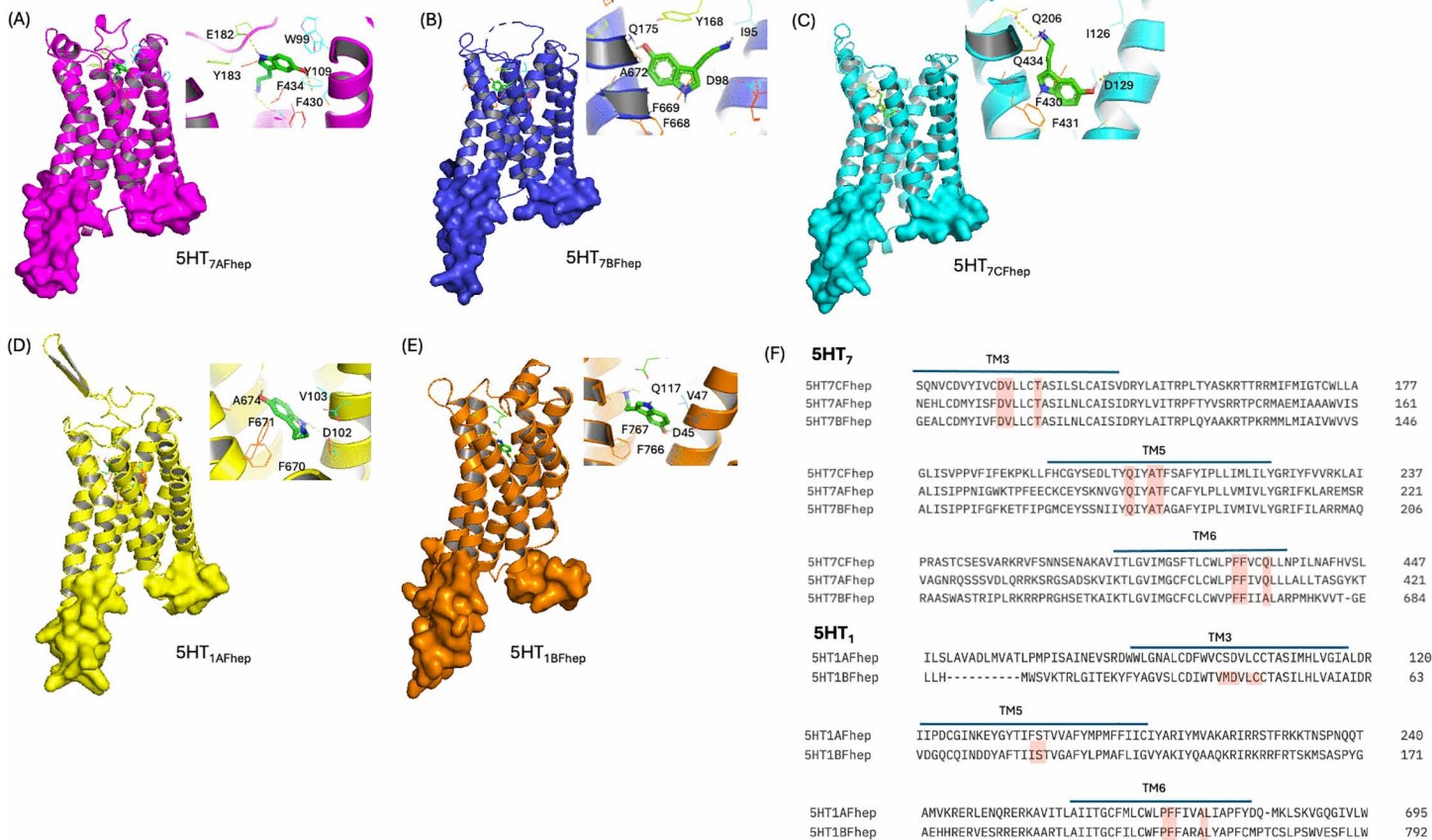

**Fig 3. Structural models of putative *Fasciola hepatica* 5-HT-gated G protein coupled receptors (GPCRs) support the presence of residues critical to serotonin binding.** (A) Predicted structural confirmation of 5HT$_{7AFhep}$ GPCR showing key ligand binding residues. (B) Predicted structural confirmation of 5HT$_{7BFhep}$ GPCR showing key ligand binding residues. (C) Predicted structural confirmation of 5HT$_{7CFhep}$ GPCR showing key ligand binding residues. (D) Predicted structural confirmation of 5HT$_{1AFhep}$ GPCR showing key ligand binding residues. (E) Predicted structural confirmation of 5HT$_{1BFhep}$ GPCR showing key ligand binding residues. All structures predicted and visualized using Phyre2 [60] and PyMOL [57] software. The serotonin (SRO) ligand was imported directly from the Protein Data Bank (https://www.rcsb.org) for binding analysis carried out by DockingPie [58]. (F) Sequence alignments of *F. hepatica* 5HT$_7$ and 5HT$_1$ receptors showing predicted essential amino acids for ligand binding. Ligand binding predicted by DockingPie [58]. Blue line highlights transmembrane regions and red shading highlights conserved amino acids predicted by docking analysis.

associated commissures at the anterior end of the parasite, consistent with the structural organization observed in other trematodes [61] (Fig 4A-4D). The IR pattern extends to the peripheral nervous system, with innervation in regions such as the pharynx and fine nerves adjacent to the surface (Fig 4A, 4B and 4D). This distribution closely resembles the serotonergic innervation pattern previously reported in adult *F. hepatica* [20], indicating that the core serotonergic nervous system is already established by the juvenile stage. These findings align with studies where serotonin has been implicated in controlling feeding and motility behaviors of related flatworms [26,33,55,62,63].

Fluorescent *in situ* hybridization (FISH) analysis demonstrated distinct spatial expression of FhTPH and 5HT$_{7CFhep}$ GPCR transcripts within the nervous system of *F. hepatica* (Fig 4E and 4F). Transcript localization (Fig 4C and 4D) identified FhTPH expression predominantly in the ventral nerve cords and associated neuronal cell bodies (Fig 4E), consistent with neuronal-based serotonin biosynthesis [64]. Similarly, 5-HT$_{7CFhep}$ GPCR expression was localized to regions within the ventral nerve cords and adjacent areas peripheral to the cells expressing FhTPH transcripts, supporting the functional coupling of serotonin synthesis and signaling via GPCRs in these regions (Fig 4F).

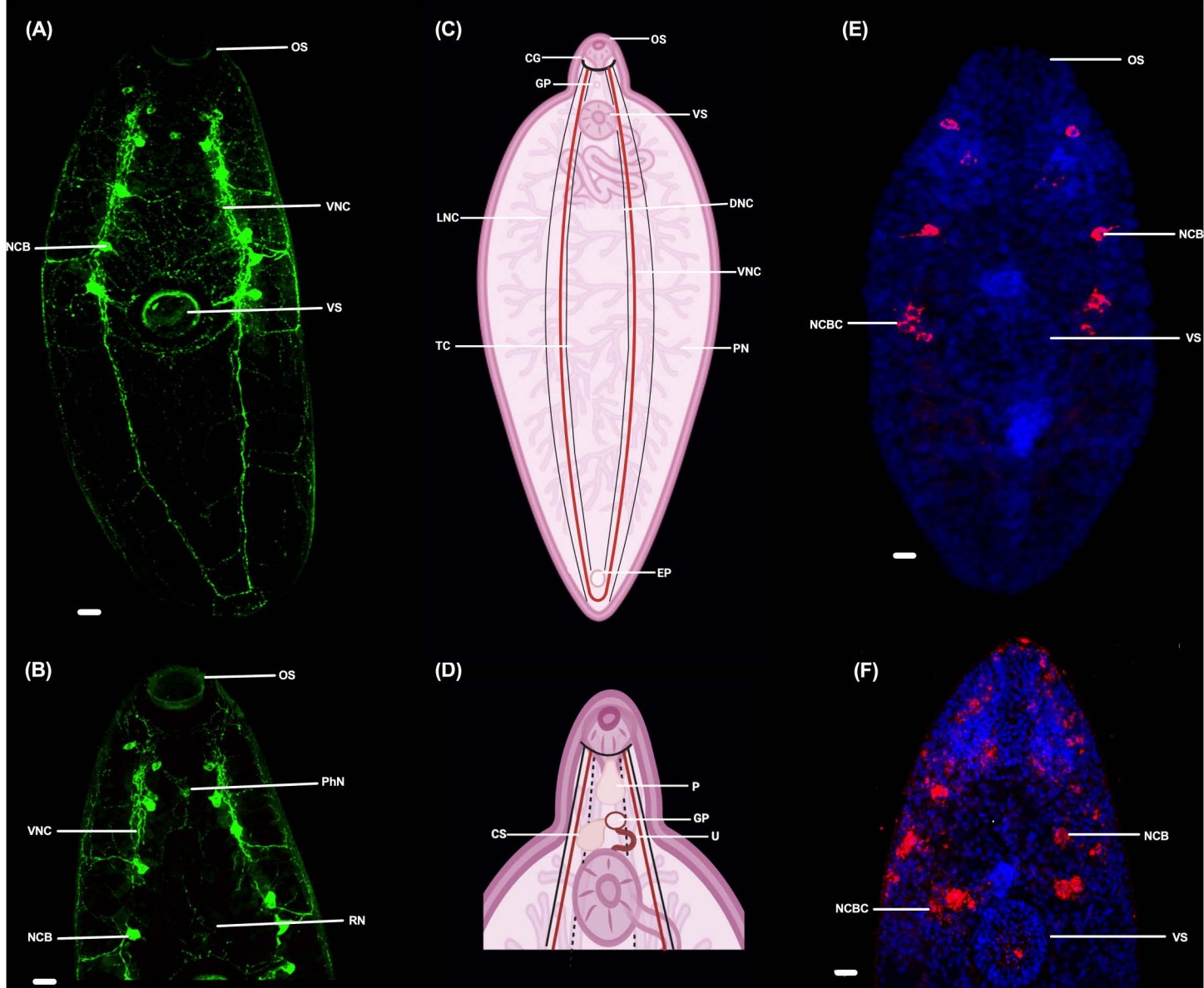

**Fig 4. Serotonin and serotonin-signaling components are widely expressed in neurons in juvenile *Fasciola hepatica*.** (A) Whole-body immunocytochemistry in 21-day-old juvenile *F. hepatica* showing widespread serotonin immunoreactivity (IR), especially in the main ventral nerve cords and brain. Green fluorescence represents serotonin IR. (B) Anterior region immunolocalization of serotonin in 21-day-old juvenile *F. hepatica*, highlighting serotonin in the cerebral ganglia and surrounding neural structures. Green fluorescence indicates serotonin IR. (C) Whole-body schematic representation of *F. hepatica* adult, illustrating key biological structures and major nervous system components, including nerve cords, commissures, and neuronal cell bodies. *Created in BioRender. McCusker, P. (2026)* https://BioRender.com/s3petk8. **(D)** Anterior schematic of F. hepatica, showing essential oral and reproductive structures, including the oral sucker, pharynx, gonopore, and cirrus sac. *Created in BioRender. McCusker, P. (2026)* https://BioRender.com/s3petk8. **(E)** Whole-body fluorescent in situ hybridization (ISH) showing *F. hepatica* tryptophan hydroxylase (FhTPH) in the nervous system. Red fluorescence indicates positive ISH signal, while blue fluorescence (DAPI staining) marks cell nuclei. FhTPH expression is observed in a pattern that aligns to neuronal cells of the nervous systems. **(F)** In situ hybridization (ISH) showing expression of the *F. hepatica* serotonin-gated G protein-coupled receptor (5HT7CFhep) in anterior neurons and adjacent regions. Red fluorescence shows receptor gene expression, while blue fluorescence (DAPI staining) highlights nuclei. Receptor localization is evident in the cerebral ganglia and other cells consistent with neuronal patterning. Abbreviations: OS = oral sucker, VS = ventral sucker, VNC = ventral nerve cord, NCB = neuronal cell body, PhN = pharyngeal nerves, RN = reproductive nerves, CG = cerebral ganglion, GP = gonopore, DNC = dorsal nerve cord, LNC = lateral nerve cord, TC = transverse commissure, PN = peripheral nerves, P = pharynx, U = uterus, CS = cirrus sac, NCBC = neuronal cell body cluster.

Both transcripts localized predominantly to the anterior end of the parasite, particularly surrounding tissues critical for host interaction and reproduction, such as the pharynx, oral suckers, and reproductive organs (Fig 4C-4F). Comparison of transcript distribution using *in situ* images indicated similar spatial expression patterns for FhTPH and 5HT$_{7CFhep}$ GPCR transcripts in juvenile fluke (S5 Fig).

### Elevated 5-HT levels enhance motility in juvenile *F. hepatica*

Exposure to 5-HT for 24 hours, in the absence of growth-enhancing media (RPMI), significantly increased the motility of *F. hepatica* newly excysted juveniles (NEJs) by over 130% at a concentration of 1 mM (Fig 5A). Shorter exposure periods (30 minutes and 4 hours) produced modest increases in motility that did not reach statistical significance and were more variable across replicates (S6 Fig), whereas sustained exposure resulted in a robust and reproducible stimulatory effect. This suggests that early stage juvenile liver fluke can uptake exogenous 5-HT, a process that has also been characterized in other flatworm species [18,33]. In *S. mansoni*, this ability has been attributed to the serotonin transporter SmSERT, which facilitates the uptake of 5-HT from the surrounding environment [18]. While higher concentrations (>1 mM) also increased motility, a significant decrease in growth and survival was observed, indicating potential toxic effects at elevated levels (S6 Fig). Dose-dependent effects of 5-HT have been reported in other parasitic species, highlighting the delicate balance required for physiological relevance [26,48].

The motility-enhancing effects of 5-HT were further corroborated by blocking serotonin reuptake from synapses using fluoxetine hydrochloride (FluHCl), a selective serotonin reuptake inhibitor (SSRI). Following a 24-hour incubation with 1 µM FluHCl, NEJ motility increased by over 170% (Fig 5A). Importantly, lower concentrations of FluHCl also elicited significantly increased motility, suggesting high affinity of this compound for the serotonin transporter (FhSERT) in *F. hepatica* (S6 Fig). This is contrary to data showing low affinity binding of fluoxetine to SmSERT in *S. mansoni* [18]. At the concentration used, the enhanced motility by fluoxetine is most consistent with increased serotonergic signaling resulting from inhibition of serotonin reuptake, although additional receptor-mediated effects cannot be excluded.

Juvenile (4 week old) *F. hepatica* motility was further enhanced by over 230% when culture media (50% CS) was supplemented with 1 mM 5-HT and maintained for 28 days (S7 Fig). This long-term exposure underscores the potential for sustained 5-HT availability to modulate neuromuscular activity in juvenile stages of *F. hepatica*. Together, these data suggest that serotonin enhances juvenile motility, consistent with a role in regulating neuromuscular functions in parasitic flatworms.

### Disruption of 5-HT signaling via RNAi induces aberrant motility in juvenile *F. hepatica*

All targeted serotonergic pathway genes were successfully knocked down by ≥50%, as confirmed by transcript quantification (S8 Fig). The transcript reduction of key 5-HT signaling pathway components had profound effects on the motility of *F. hepatica* juveniles, underscoring the role of 5-HT in neuromuscular coordination. Knockdown of the 5-HT synthesis enzyme, FhTPH, resulted in a significant 84% reduction in juvenile motility, demonstrating that endogenous serotonin synthesis is essential for maintaining motor activity. In contrast, silencing FhAADC1 and FhAADC2 did not significantly affect fluke motility, suggesting these enzymes are not rate limiting and/or there are unknown compensatory mechanisms within endogenous monoaminergic pathways systems, consistent with the additional role of AADC enzymes in dopamine biosynthesis [15]. Knockdown of the vesicular monoamine transporter, FhVMAT, responsible for serotonin storage and release, caused a significant 58.5% reduction in juvenile motility (Fig 5B). This supports the hypothesis that synaptic availability of serotonin is required for normal locomotor output. While TPH and VMAT have been identified in other parasites [15,39], this study provides the first characterization of their functional importance in a parasitic flatworm. In contrast, knockdown of the 5-HT reuptake transporter, FhSERT, likely to impede serotonin removal from synaptic junctions, led to a dramatic 235% increase in juvenile motility (Fig 5B), providing the first SERT-associated phenotype in a parasitic flatworm and functional evidence that serotonin clearance limits 5-HT signaling in *F. hepatica*.

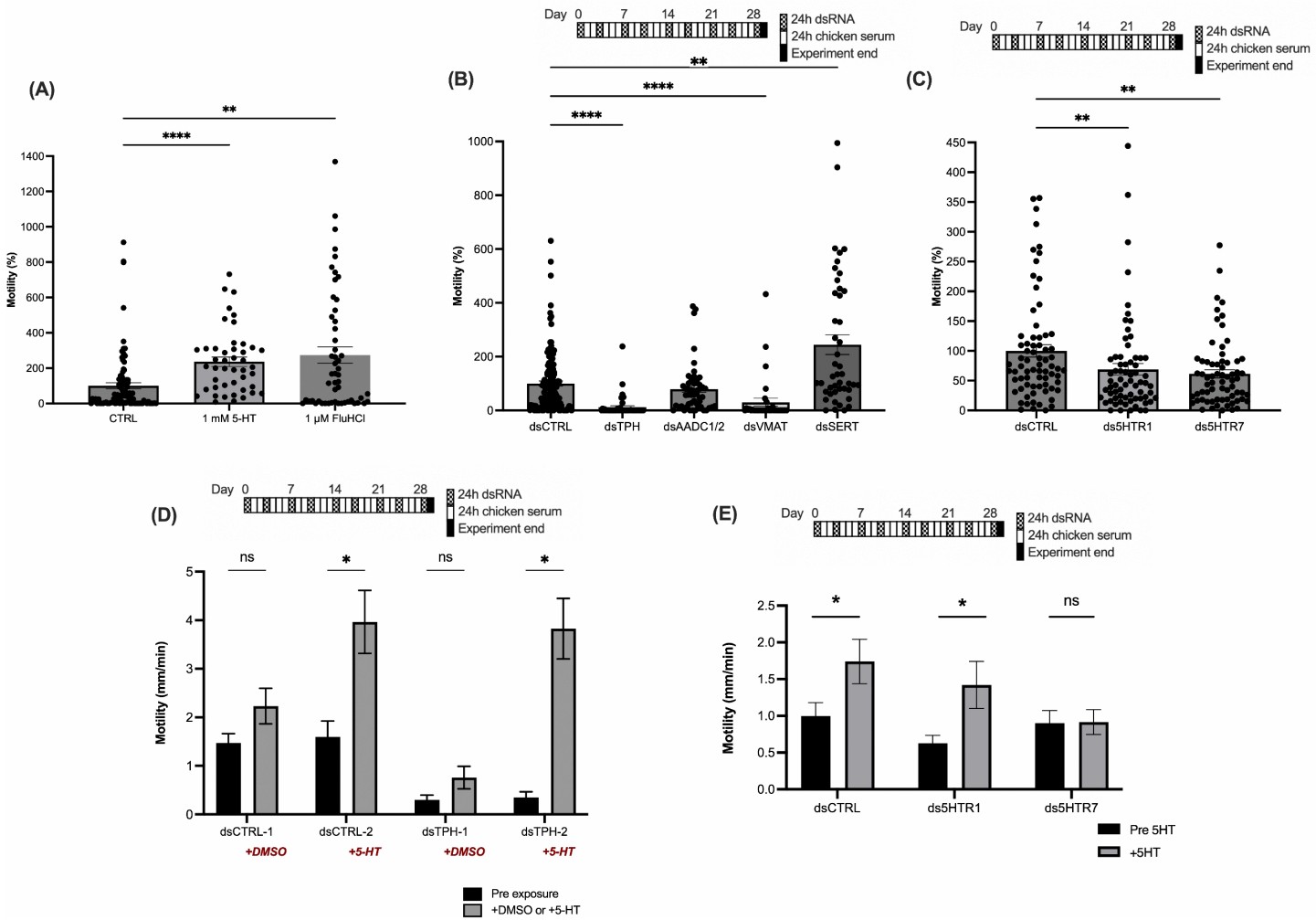

**Fig 5. Chemical inhibition and gene-silencing of serotonin-signaling components in juvenile *Fasciola hepatica* alter motility.** (A) Increased motility in newly excysted juvenile *F. hepatica* following 24-hour exposure to 1 mM serotonin (5-HT) and 1 µM fluoxetine hydrochloride (FluHCL). Each data point represents motility (percentage length change per mm relative to the negative control) of an individual worm. Statistical significance determined by ANOVA; **$p \leq 0.01$, ****$p \leq 0.0001$; minimum $n = 45$ NEJs per treatment group. (B) Altered motility of 28-day-old juvenile *F. hepatica* after RNAi knockdown of serotonin-signaling pathway genes. Targeted genes include FhTPH (tryptophan hydroxylase), FhAADC1 and FhAADC2 (aromatic amino acid decarboxylase enzymes), FhVMAT (vesicular monoamine transporter), and FhSERT (serotonin reuptake transporter). Motility is expressed as percentage movement (mm/min) of individual juveniles compared to untreated controls. dsRNA exposure carried out twice per week as shown. Statistical significance determined by ANOVA; **$p \leq 0.01$, ****$p \leq 0.0001$; minimum $n = 36$ juveniles per treatment group. (C) Decreased motility of 28-day-old juvenile *F. hepatica* following combinatorial RNAi knockdown of serotonin-G protein-coupled receptors (GPCRs). Targeted receptors include type 1 (5HTR1AFhep and 5HTR1BFhep) and type 7 (5HTR7AFhep, 5HTR7BFhep, and 5HTR7CFhep) serotonin receptors. Data points represent motility (percentage, mm/min) of individual juveniles compared to untreated controls. dsRNA carried out twice per week as shown. Statistical significance determined by ANOVA; **$p \leq 0.01$.; minimum $n = 68$ juveniles per treatment group. (D) Serotonin rescues the motility impact of FhTPH-knockdown in *F. hepatica* juveniles. Black bars represent basal motility (mm/min) of juveniles post repeated double stranded (ds)RNA exposure (dsCTRL or dsTPH) but pre-chemical exposure. Gray bars show basal motility (mm/min) of juveniles post-dsRNA treatment followed by 18-hour exposure to 0.001% dimethyl sulfoxide (DMSO) or 1 mM 5-HT as outlined in red. Statistical significance determined by t-test; *$p \leq 0.05$; minimum $n = 30$ juveniles per treatment group. (E) The silencing of Fh5HT type 7 GPCRs abolishes the stimulatory effects of 1 mM 5-HT on juvenile liver fluke. Black bars represent percentage motility (compared to RPMI) of juveniles post repeated dsRNA exposure (dsCTRL, ds5HTR1 [5HTR1AFhep, 5HTR1BFhep] or ds5HTR7 [5HTR7AFhep, 5HTR7BFhep, and 5HTR7Cfhep]) but pre-chemical exposure. Gray bars show motility after dsRNA treatment followed by 18-hour exposure to 1 mM 5-HT. Statistical significance determined by t-test; *$p \leq 0.05$; minimum $n = 30$ juveniles per treatment group.

To assess receptor-specific contributions to motility, receptor clades were targeted using RNAi in a combinatorial RNAi approach. Simultaneous knockdown of type 1 GPCRs using dsRNAs targeting both Fh5HT$_{1AFhep}$ and Fh5HT$_{1BFhep}$ reduced motility by ~32%, while knockdown of type 7 GPCRs using dsRNAs targeting Fh5HT$_{7AFhep}$, Fh5HT$_{7BFhep}$, and Fh5HT$_{7CFhep}$ receptors produced a~39% reduction in motility (Fig 5C). These findings support the hypothesis that 5-HT GPCRs play a role in modulating neuromuscular activity in juvenile *F. hepatica*.

## Exogenous 5-HT rescues RNAi-induced motility defects

Knockdown of FhTPH significantly reduced the motility of juvenile *F. hepatica*, consistent with an impaired ability to synthesize 5-HT endogenously. To validate that reduced motility could be functionally rescued by restoring serotonergic signaling, exogenous 5-HT was applied to FhTPH-silenced juveniles. Exposure to 1 mM 5-HT for 18 hours restored and enhanced motility of FhTPH-silenced juveniles by tenfold, confirming that the phenotype reflects impaired serotonin availability rather than irreversible dysfunction (Fig 5D). This validated rescue approach was applied in a separate experimental setup to assess the functional roles of 5-HT receptors in juvenile motility. Exogenous 5-HT increased motility and partially rescued the motility deficits of juveniles with silenced 5HT$_1$ receptors (Fig 5E), indicating that these worms retained the capacity to respond to elevated serotonin levels. In contrast, no rescue was observed in juveniles with silenced 5HT$_7$ receptors (Fig 5E), despite clear RNAi-induced motility deficits. Although knockdown of both 5HT$_1$ and 5HT$_7$ receptors significantly reduced motility in the initial receptor RNAi experiment (Fig 5C), the failure of exogenous 5-HT to restore movement following 5HT$_7$ silencing provides strong functional evidence that 5HT$_7$ receptors are required for the enhanced motility effects of serotonin under the conditions tested. This interpretation is consistent with the functional characterization of the *S. mansoni* 5-HT$_7$ receptor, where RNAi-mediated silencing resulted in a marked reduction in schistosomula motility, underscoring the receptor's critical role in movement regulation [26]. The reduced motility observed following 5HT$_1$ knockdown could reflect broader or indirect physiological roles, rather than direct mediation of acute serotonergic excitation. Although 5HT$_1$ receptors clearly contribute to motility regulation, their role may be more context-dependent. In vertebrates, 5HT$_1$ receptors are commonly G$_{i/o}$-coupled and involved in diverse modulatory functions within neural circuits [46,65]; whether comparable signaling roles operate in parasitic flatworms remains to be determined.[65]

## Elevated 5-HT levels enhance growth of juvenile *F. hepatica*

Interestingly, juveniles maintained in a medium consisting of 50% chicken serum supplemented with 1 mM 5-HT over a 28-day period exhibited a statistically significant 33% increase in growth (Fig 6A and 6B). This growth response was variable across individuals, with a subset of treated parasites exhibiting more pronounced increases in size, but the overall effect was significant when assessed across the full dataset. These findings suggests that elevated 5-HT levels can influence growth trajectories of juvenile *F. hepatica* under *in vitro* culture conditions. 5-HT is well-documented as a critical neuromodulator in invertebrates, with roles extending beyond neurotransmission to include the regulation of growth, reproduction, and metabolism. In parasitic flatworms, previous studies have highlighted serotonin's role in modulating motility and host-parasite interactions, but its influence on growth has been less explored. The present data therefore extend existing knowledge by indicating that serotonergic signaling may contribute to growth regulation among individual fluke juveniles. There are a few studies that suggest 5-HT may influence cell proliferation and tissue growth through its influence on energy metabolism and cell division [28,29,31,32]. 5-HT stimulates glucose uptake and promotes glycogen breakdown in *F. hepatica* and *S. mansoni*, leading to increased lactic acid production under anaerobic conditions [28,29]. This metabolic shift suggests that 5-HT enhances glycolysis, ensuring a rapid energy supply necessary for energy-intensive processes such as motility. This metabolic support may also facilitate growth in a proportion of *F. hepatica* juveniles, particularly under nutrient-rich *in vitro* conditions. Emerging research suggests that 5-HT plays a pivotal role in directly regulating cell proliferative mechanisms across a range of species. Although evidence remains limited in invertebrates, studies in *E. multilocularis* have demonstrated that 5-HT signaling is essential for larval development and

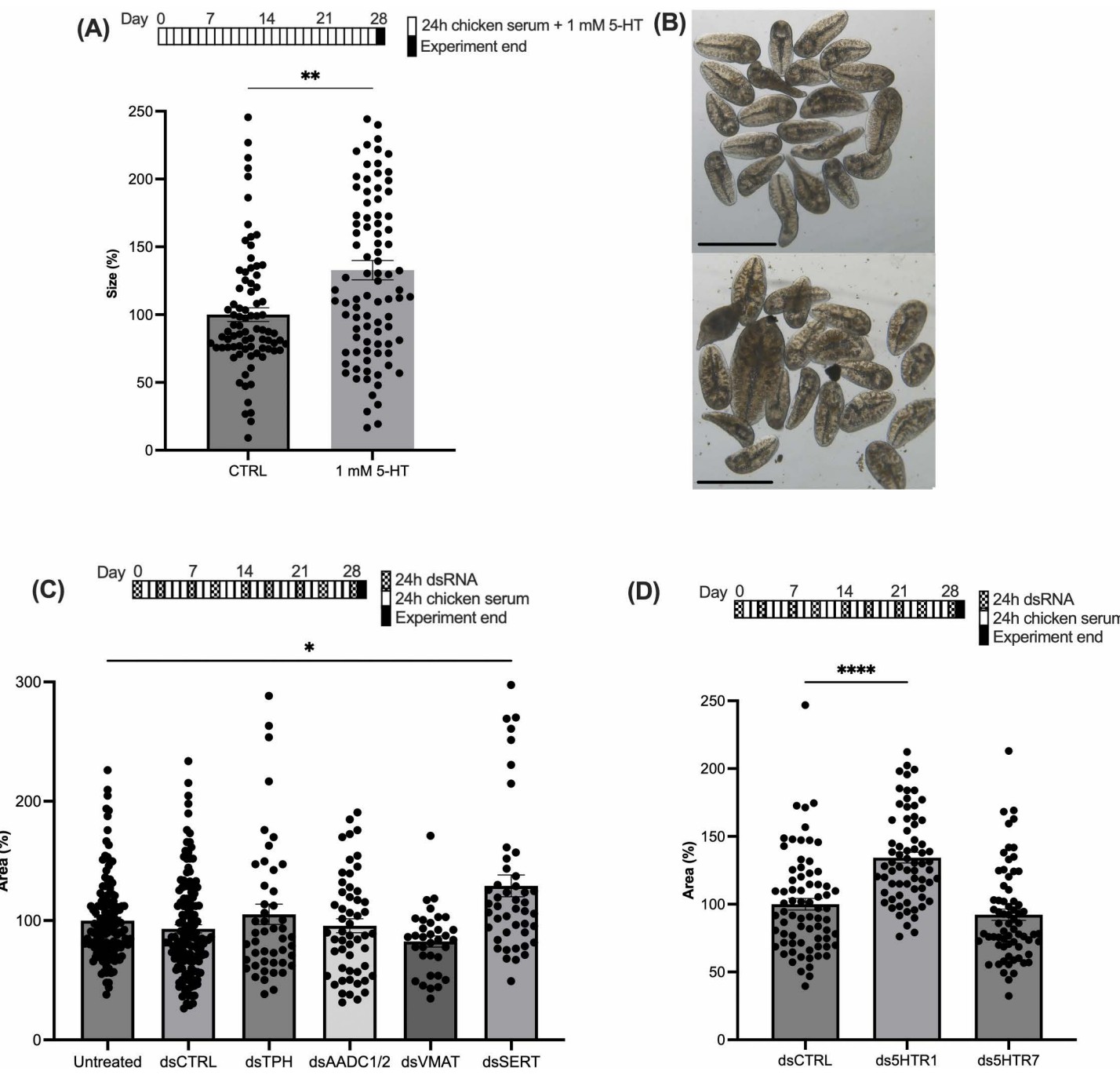

**Fig 6. Serotonin and the gene silencing of selected serotonin-signaling components alter juvenile *Fasciola hepatica* growth.** (A) The growth enhancing effect of prolonged exposure (28 days) to 1 mM serotonin (5-HT) on juvenile *F. hepatica*. Each data point represents the size of an individual juvenile expressed as percentage area (mm²) relative to the unsupplemented media control. Statistical significance determined by t-test analysis; **p ≤ 0.01; minimum n = 80 juveniles per treatment group. (B) Brightfield images showing the growth enhancing impact of 1 mM 5-HT supplementation on 21-day-old *F. hepatica* juveniles. Top image shows snapshot of 21 randomly selected juveniles from 80 total cultured juveniles without 5-HT supplementation, bottom image shows snapshot of 19 randomly selected juveniles from 91 total cultured juveniles with 5-HT supplementation. (C) Growth enhancing effect of FhSERT silencing in 28-day-old *F. hepatica* juveniles. Targeted genes include FhTPH (tryptophan hydroxylase), FhAADC1 and FhAADC2 (aromatic amino acid decarboxylases), FhVMAT (vesicular monoamine transporter), and FhSERT (serotonin reuptake transporter). Growth is expressed as percentage area (mm²) of individual juveniles compared to untreated controls. dsRNA exposure carried out twice a week as shown.

Statistical significance determined by ANOVA; *p ≤ 0.05; minimum n = 34 juveniles per treatment group. (D) RNAi-mediated knockdown of type 1 serotonin-G protein-coupled receptors (GPCRs) in 28-day-old juvenile *F. hepatica* enhanced growth. Targeted receptors include type 1 (5HTR1AFhep and 5HTR1BFhep) and type 7 (5HTR7AFhep, 5HTR7BFhep, and 5HTR7CFhep). Each data point represents the relative area (percentage, mm²) of an individual juvenile compared to untreated controls. dsRNA exposure carried out twice a week as shown. Statistical significance determined by ANOVA; ****p ≤ 0.0001; minimum n = 64 juveniles per treatment group.

promotes the expansion of germinal layer cells, which are critical for parasite growth and survival [33]. In vertebrates, the role of 5-HT in cell proliferation is more extensively documented, particularly in the context of cancer – for recent review see [66]. For example, in the central nervous system, 5-HT modulates neurogenesis by regulating both the proliferation and differentiation of neural progenitor cells [67,68], whilst treating non-small cell lung cancer cells with 5-HT enhanced their proliferation and migration [69]. Here, the growth-enhancing effect of 5-HT was substantiated for *F. hepatica,* as the growth of juveniles increased significantly by 29.1% following RNAi-mediated knockdown of the 5-HT transporter gene (FhSERT) (Fig 6C). No significant change in growth was observed with knockdown of other 5-HT pathway genes, highlighting the importance of 5-HT transport in modulating serotonergic signaling and its downstream effects on parasite biology. RNAi-mediated knockdown of GPCRs provides further evidence of the role of 5-HT in developmental processes, particularly through its interaction with $5HT_1$ receptors. A significant 22.3% increase in juvenile size was observed when $5HT_1$ receptors were silenced (Fig 6D), suggesting $5HT_1$ receptors may function as inhibitory regulators of growth, where their silencing removes this inhibition, thus enhancing growth. While the precise pathways of action remain under investigation, existing research suggests that, beyond its well-documented role in inhibiting cAMP production, 5-HT is directly involved in regulating multiple cell cycle progression pathways [66]. Notably, it has been linked to the mitogen-activated protein kinase (MAPK) and PI3K/Akt signaling cascades, which are essential for cellular growth, survival and differentiation [70], as well as the Hippo signaling pathway, a key regulator of organ size and cell proliferation [71]. These interactions are thought to be mediated by specific 5-HT receptor subtypes, particularly $5HT_7$ and $5HT_1$ receptors [72,73].

The interplay between serotonin receptor subtypes in regulating developmental processes in fluke underscores the complexity of 5-HT signaling. Future research should aim to elucidate the precise molecular mechanisms underlying 5-HT-mediated growth in juvenile liver fluke, with a particular focus on identifying downstream pathways involved in cell proliferation and tissue differentiation. A deeper understanding of these processes could significantly expand the repertoire of drug targets associated with serotonin signaling.

## Conclusion

The findings of this study provide compelling evidence that serotonin (5-HT) plays a pivotal role in regulating not only neuromuscular function, but also growth and development in *F. hepatica*. These findings establish serotonin as a fundamental regulator of *F. hepatica* biology, offering new insights into its potential as a therapeutic target for parasite control.

## Methods

### Computational characterization of serotonergic signaling pathway of *F. hepatica*

Key genes involved in serotonergic signaling were identified using BLAST analyses of the *Fasciola hepatica* predicted protein datasets (PRJEB58756) with characterized human (TPH; P17752, AADC; P20711, VMAT; Q05940, SERT P31645) and *Schistosoma mansoni* (TPH; Smp_174920, AADC; Smp_171580, VMAT; Smp_121920, SERT; Smp_333840) orthologs as queries [15,18,35–39,74]. Predicted protein datasets used in this study were downloaded from WormBase ParaSite (version WBPS19; https://parasite.wormbase.org/index.html) [75]. Annotated GPCR sequences [15,26,40,54] were used to interrogate the genomes of *F. hepatica* and 28 other flatworm species (16 cestodes; *Echinococcus canadensis* (PRJEB8992), *Echinococcus granulosus* (PRJNA182977 - G1), *Echinococcus multilocularis* (PRJEB122 - Java), *Echinococcus oligarthrus* (PRJEB31222 - DaMi1), *Hymenolepis diminuta* (PRJEB507 - Denmark), *Hymenolepis microstoma*

(PRJEB124), *Hymenolepis nana* (PRJEB508 - Japan), *Mesocestoides corti* (PRJEB510 - Specht & Voge 1965), *Taenia asiatica* (PRJEB532 - South Korea), *Taenia multiceps* (PRJNA307624 - Gns01), *Taenia saginata* (PRJNA71493 - TSAYD01), *Taenia solium* (PRJNA170813 - Mexico), *Dibothriocephalus latus* (PRJEB1206), *Spirometra erinaceieuropaei* (PRJEB1202), *Schistocephalus solidus* (PRJEB527 - NST_G2), *Hydatigera taeniaeformis* (PRJEB534 - Spain/Canary Islands) and 12 trematode species; *Clonorchis sinensis* (PRJNA386618 - Cs-k2), *Schistosoma bovis* (PRJNA451066), *Schistosoma japonicum* (PRJNA520774 - HuSjv2), *Fasciolopsis buski* (PRJNA284521 - HT), *Fasciola gigantica* (PRJNA230515 - Uganda_cow_1), *Schistosoma haematobium* (PRJNA78265), *Schistosoma curassoni* (PRJEB519 - Senegal/Dakar), *Schistosoma mansoni* (PRJEA36577), *Trichobilharzia regenti* (PRJEB44434 - tdTriRege1.1), *Opisthorchis viverrini* (PRJNA222628), *Paragonimus westermani* (PRJNA454344), *Echinostoma caproni* (PRJEB1207 - Egypt)) using Hidden Markov Models (HMM) to identify additional 5-HT receptor candidates [76]. Protein hits with E-values <0.01 were validated through reciprocal BLAST searches against National Center for Biotechnology Information (NCBI) non-redundant (nr) database sans Platyhelminthes and UniProt datasets [74,77]. GPCRs with ≥4 transmembrane domains were confirmed using TMHMM-2.0 [76], InterProScan (version 5; [78]), and conserved motif analysis. Expression levels were quantified as transcripts per million (TPM) from published datasets [44,75]. Alignments were performed with Clustal Omega [79], and maximum likelihood phylogenetic trees were generated using MEGA (version 11; [80]) (model = WAG + F; bootstrap = 500) and formatted using iTOL [81] Protein relationships were further visualized through CLANs analysis [82]. GPCR structures were predicted using Phyre2.2 software [60] and imported into PyMOL [57] for visualization and ligand binding analysis using the PyMOL plugin DockingPie 1.2 [58]. The serotonin (SRO) ligand structure was downloaded from Protein Data Bank (PDB) and imported directly into PyMOL for docking analysis [83].

### *F. hepatica* and excystment

Juvenile *F. hepatica* of Italian strain (obtained from Ridgeway Research) were prepared and maintained as follows: metacercariae were stored in distilled water at 4°C and used within 4 months of receipt. Over this period, excystment efficiency was consistently >70%. Metacercariae were excysted following the protocol described by McVeigh et al. [84]. Prior to excystment, metacercariae were manually popped from their outer wall casing and bleached in 10% sodium hypochlorite solution. 'Newly excysted juveniles (NEJs)' are defined as juveniles excysted within 24 hours of experimentation, whilst later stage 'juveniles' denote life stages that have been further cultured to developing fluke that are transcriptionally distinct from NEJs and stage matched ex *in vivo* juveniles as described by Robb et al., 2022 [41]. Juveniles are cultured at 37°C in a humidified atmosphere with 5% $CO_2$, either in 50:50 chicken serum:RPMI 1640 medium (CS: Merck, C5405; RPMI 1640; Thermo Fisher Scientific) or in RPMI 1640 alone, depending on the experimental application, as outlined by McCusker et al. [85]. The culture medium was replaced three times per week during extended culture periods. The complete protocol can be accessed at: dx.doi.org/10.17504/protocols.io.14egn212qg5d/v1

### Immunocytochemistry

Fifteen- twenty 21 day old juveniles were flat fixed under a coverslip for 10 minutes at room temperature and free fixed overnight at 4 °C in 4% paraformaldehyde. Fixed juveniles were washed and stored in antibody diluent (AbD; 0.1 M phosphate-buffered saline (PBS) (Merck), 0.1% Triton-X (Merck), 0.1% bovine serum albumin (Merck)) at 4 °C until required. Serotonin primary antibody (S5545, Merck) incubations were carried out at 1:1000 dilution in AbD for 72 hours rotating at 4 °C. Juveniles were washed in AbD before incubation in fluorescein isothiocyanate (FITC)-labelled anti-rabbit secondary antiserum (Merck) at 1:100 concentration for 48 hours rotating at 4 °C. Juveniles were further washed in AbD before mounting in Vectashield (Vector Laboratories) and imaged on a Leica TCS SP8 confocal scanning laser microscope.

### Whole mount fluorescence *in situ* hybridization (FISH)

*In situ* hybridization methods were carried out as described in detail by Armstrong et al. [86] In summary, after 4 weeks of culture, 20–25 juveniles per probe were fixed under a coverslip in 4% formaldehyde for 10 minutes, followed by free

fixation on a rotator for another 10 minutes at room temperature. They were then washed in PBSTx (1×PBS with 0.3% Triton-X) and dehydrated in 50% methanol:PBSTx before storage in 100% methanol at −20°C. RNA probes for FhTPH and $5HT_{7CFhep}$ were synthesized from T7 templates amplified via PCR using the Roche FastStart kit. Probe primers are outlined in S4 Table. Probes targeting multiple serotonergic GPCRs were designed and tested; however, consistent and reproducible FISH signal was obtained only for the $5HT_{7cFhep}$ receptor. Probe synthesis involved overnight incubation at 27°C with DIG-UTP labelling mix (Roche), transcription buffer, and T7 RNA polymerase (T7 RNA polymerase kit, ThermoFisher Scientific), followed by DNase treatment (0.02%) and ethanol precipitation with 0.1 M lithium chloride at −80°C. RNA pellets were centrifuged (x 16000$g$, 4 °C), washed in 70% ethanol, resuspended in nuclease-free water, and assessed for purity and concentration using a DeNovix DS-11 FX spectrophotometer. Probes were diluted to 50 ng/μL in hybridization buffer

(50% de-ionized formamide, 25% saline-sodium citrate buffer (SSC), 0.1 mg/ml yeast RNA, 1% (v/v) Tween-20, 5% dextran sulfate, made to volume with DEPC-treated $H_2O$) and stored at -20 °C. Yeast RNA was prepared per Jing [87].

*In situ* hybridization methods were adapted from methods of *Schistosoma mansoni* and *Schmidtea mediterranea* [88]. A full protocol is available at https://www.protocols.io/view/fluorescent-in-situ-hybridisation-for-juvenile-fas-3byl4q7x-jvo5/v1. Flat-fixed juveniles were rehydrated with 10-minute washes in 50% PBSTx:MeOH and 1×SSC, then bleached (8.85 ml $H_2O$, 0.5 ml formamide, 0.25 ml 20x SSC, 0.4 ml 30% $H_2O_2$) under light for 1.5 hours. Worms were further washed for 10 minutes in 1x SSC and PBSTx. Juveniles were permeabilized with proteinase K solution (0.1% SDS and 0.01 mg/ml proteinase K) and post-fixed for 10 minutes in 4% formaldehyde. Worms were prepared into Intavis baskets (35 μm mesh) and incubated for 10 minutes in PBST:Pre-hybridization solution (50% de-ionized formamide, 25% SSC, 0.1 mg/ml yeast RNA, 1% (v/v) Tween-20, made up to volume with DEPC-treated $H_2O$) before a 2-hour incubation at 52 °C in pre-hybridization solution. Probes were pre-heated at 80°C for 5 minutes and hybridized with juveniles overnight (≥16 h) at 52°C using 1 ng/μl probe solution; sense probes served as controls. Post-hybridization, worms were washed in 50% hybridization solution: 2x SSCx (2x SSC with 0.1% Triton X-100), 2x SSCx and 0.2x SSCx at 52 °C, followed by two 10-minute TNTx (0.1 M Tris pH 7.5, 0.15 M NaCl, 0.1% Tween-20) washes at room temperature. Blocking was performed with 5% inactivated horse serum and 0.5% Western Blocking Reagent in TNTx for 2 hours, followed by overnight incubation at 4°C with 1:2000 anti-DIG-POD (diluted in blocking solution, Merck). Worms underwent sequential TNTx washes (5, 10 and 6x 20 minutes) at room temperature and were incubated in tyramide solution (TSA buffer; 2 M NaCl; 0.1 M Boric acid pH 8.5, 0.5% $H_2O_2$, 0.1% 4-Iodophenylboronic acid (4IPBA) 20 mg/ml, 1:500 5-TAMRA (5-carboxytetramethylrhodamine, Merck) for 10 minutes at room temperature. Worms were stained with 4,6-diamidino-2-phenylindole (DAPI) overnight and washed in TNTx for 48 hours. Juveniles were mounted in Vectashield before imaging using a Leica TCS SP8 confocal microscope.

## Chemical exposures

Serotonin hydrochloride powder (H9523, Merck) was dissolved to 100 mM stock concentration in MilliQ water. For short term exposures, juveniles were exposed to a final concentration of 0.01 μM – 1 mM serotonin in 3 ml RPMI. For long exposures, juveniles were cultured in 200 μl 50% CS supplemented with 0.01 μM – 1 mM serotonin. For each treatment group, a minimum of 10 juveniles were used per replicate, with at least three independent biological replicates performed. All raw data are provided in S3 Table. Basal serotonin concentrations in RPMI and CS media are not routinely provided by manufacturers, therefore, serotonin effects were quantified relative to matched negative controls (RPMI-only for short-term exposures; RPMI+50% CS for long-term exposures). Vehicle controls were also included as appropriate; dimethyl sulfoxide (DMSO) in dd$H_2O$ at the same final dilution used for serotonin stocks. Fluoxetine hydrochloride was dissolved to 10 mM stock concentration in DMSO and further diluted to desired final concentrations (0.01 μM – 1 mM) in 3 ml RPMI 1640. 0.02% dd$H_2O$ and 0.001% DMSO were used as negative controls as required. Images and videos were captured using an Olympus SZX10 stereo microscope with CAM-SC50 camera to monitor phenotypic changes (area- mm$^2$; motility-length change, mm/min) determined via ImageJ analysis using WrMTrcK plugin [89,90].

## RNA interference

*F. hepatica* juvenile RNAi was carried out as described by McCusker et al. [91]. Double stranded RNAs (dsRNA) specific to bacterial neomycin (bacterial neomycin phosphotransferase, U55762), red fluorescent protein (cloning vector pAJ50, ARW80046), *F. hepatica* tryptophan hydroxylase (FhTPH; FhHiC23_g11603), *F. hepatica* aromatic decarboxylase 1&2 (FhAADC1; FhHiC23_g50, FhAADC2; FhHiC23_g2511), *F. hepatica* vesicular monoamine transporter (FhVMAT; FhHiC23_g2), *F. hepatica* serotonin transporter (FhSERT; FhHiC23_g1585) and five serotonin receptors (FhHiC23_g4917, FhHiC23_g1196, FhHiC23_g115, FhHiC23_g2340, FhHiC23_g7880) were generated using the T7 RiboMAX Express RNAi System (Promega) from cDNA templates labelled with the T7 promoter sequence; 5'-TAATACGACTCACTATAGGGT-3' amplified by PCR. All primers are listed in S4 Table. All RNAi experiments were performed with a minimum of three independent biological replicates, with a minimum of 10 juveniles used per treatment group in each replicate. Juveniles were exposed to 100 ng/µl dsRNA (target or control) in RPMI 1640 for 24 hours twice per week for 4 weeks under normal culture conditions. Between dsRNA exposures juveniles were cultured as standard in 50% CS. Post RNAi exposures, juveniles were snap frozen for extraction and transcript quantification. Phenotypic analysis was carried out at day 29 post excystment after dsRNA exposure, comparing to RPMI and dsRNA control groups.

## Transcript analysis

mRNA was extracted from RNAi treatment groups using Dynabeads mRNA direct Kit (Thermo Fisher Scientific). cDNA was synthesized (High Capacity RNA-to-cDNA kit, Thermo Fisher Scientific) following DNase treatment (Turbo DNA-free, Thermo Fisher Scientific). qPCRs were performed on a Rotor-Gene Q 5-plex HRM PCR system (Qiagen) using SensiFast SYBR (Bioline) and primers outlined in S4 Table at final concentration 1.25 µM. Cycling parameters; 10 minutes at 95 °C, followed by 40 cycles of 95 °C 10 s, 60 °C 15 s, 72 °C 30 s. All PCRs were performed in triplicate and included no-template controls and melt-curve analyses as standard. Relative expression analysis was determined using Pfaffl's Augmented ΔΔCt method [92], normalizing expression in each sample relative to the untreated control, standardized to a glyceraldehyde 3-phosphate dehydrogenase (GAPDH, FhHiC23_g5828) housekeeping gene. Statistical significance was determined relative to the effects of negative control treatments (dsCTRL) on target gene expression.

## Statistical analysis

Growth and motility data were analyzed using one-way ANOVA with Dunnett's post hoc test, analyzing significance of treatments relative to the negative control dsRNA treated juveniles (RNAi experiments), or vehicle control treated juveniles (compound exposure experiments). All raw data for figures can be found in S3 Table [20].

## Supporting information

**S1 Fig. Alignment of flatworm serotonin (5-HT) gated G protein coupled receptors. (A) Alignment of transmembrane 3 region.** Red box highlighting conserved serotonin-binding motif 'DVXXCT' essential for ligand recognition, as well as the canonical 'DRY' motif critical for GPCR activation. **(B) Alignment of transmembrane 5 region.** Red box highlighting conserved serotonin-binding motif 'WXXF' essential for ligand recognition.
(DOCX)

**S1 Table. Summary of bioinformatic analysis defining serotonin signaling genes in *Fasciola hepatica.*** Gene IDs generated from interrogation of *F. hepatica* genome annotation PRJEB58756 (WormBase ParaSite version 19). Data shown; reciprocal BLAST (NCBI nr and Uniprot), Interproscan domain analysis and transcript per million (TPM) data [44,74,75,77,78].
(XLSX)

**S2 Fig. Motif analysis of *Fasciola hepatica* serotonin receptors.** Transmembrane 3 (TM3); red boxes show Tyrosine (Y) or Tryptophan (W) residues consistent with $5HT_7$ or $5HT_1$ receptors and Cysteine (C) and Threonine (T) residues consistent with all $5HT_1$ and $5HT_7$ receptors [48]. TM5; red boxes show 'AXXXXA' pattern of Alanine (A) residues consistent with $5HT_7$ receptor classification or 'SXXXXA' pattern of Serine (A) and Alanine (A) residues consistent with $5HT_1$ receptor classification [48].
(DOCX)

**S2 Table. Flatworm serotonin (5-HT) gated G protein coupled receptor gene sequences.** Dataset generated by BLAST analysis against flatworm genomes on WormBase ParaSite (WBPSv19), reciprocal BLAST outputs shown from Uniprot database and domain analysis from Interproscan [74,75,77,78].
(XLSX)

**S3 Fig. G protein coupling predictions for *Fasciola hepatica* serotonin G protein coupled receptors.** Analysis carried out using PRED COUPLE 2.0 software [56], positive coupling determined by default 0.3 threshold.
(DOCX)

**S3 Table. Raw data for manuscript Figs 5 and 6.** Data includes percentage motility and growth values (%) as presented on graphs, raw area values ($mm^2$) for growth analysis and percentage survival (%) of juveniles across total experiment length.
(XLSX)

**S4 Fig. (A)** PyMOL model of human $5HT_1$ G protein coupled receptor (GPCR; P08908). **(B)** PyMOL model of human $5HT_7$ GPCR (P34969). **(C)** Alignment of human $5HT_1$ GPCR and *F. hepatica* 5-HT GPCRs ($5HT_{1AFhep}$ and $5HT_{1BFhep}$). Root mean square deviation (RMSD) values show structural similarity with human $5HT_1$ GPCR (P08908). **(D)** Alignment of human $5HT_7$ GPCR and *F. hepatica* 5-HT GPCRs ($5HT_{7AFhep}$, $5HT_{7BFhep}$ and $5HT_{7CFhep}$). Root mean square deviation (RMSD) values show structural similarity with human $5HT_7$ 5-HT GPCR (P34969).
(DOCX)

**S4 Table. Primer details for double stranded (ds)RNA generation, real time (q)PCR amplification and fluorescent in situ hybridisation (FISH) of targets from *Fasciola hepatica* serotonin pathway.** T7 primer, TAATACGACTCAC-TATAG, added to generate dsRNA and FISH probe amplicons.
(XLSX)

**S5 Fig. Fluorescent in situ hybridisation of *Fasciola hepatica* tryptophan hydroxylase (FhHiC23_g11603) and G protein coupled receptor (FhHiC23_g2340).** Red boxes highlight similar expression patterns and highlight key neuronal cells expressing serotonin signalling genes.
(DOCX)

**S6 Fig. (A)** Motility of newly excysted juveniles (NEJs) after 30 minute exposure and 4 hour exposure to 1 mM serotonin (5-HT). Dots represent motility (mm/min) of individual juveniles. Significance determined via one-way ANOVA analysis and Dunnett's multiple comparisons test; ns = no significance. **(B)** Motility of 21 day old *Fasciola hepatica* after 24 hour exposure to 1 mM, 3 mM, 5 mM or 10 mM 5-HT. Dots represent motility (mm/min) of individual juveniles. Significance determined via one-way ANOVA analysis and Dunnett's multiple comparisons test; *** $p \leq 0.005$, **** $p \leq 0.001$. **(C)** Area of 21 day old *F. hepatica* after incubation in 50% chicken serum supplemented with 1 mM, 3 mM, 5 mM or 10 mM 5-HT. Area = $mm^2$. Significance determined via one-way ANOVA analysis and Dunnett's multiple comparisons test; * $p \leq 0.05$. **(D)** Survival of juvenile *F. hepatica* after incubation in 50% chicken serum supplemented with 1 mM, 3 mM, 5 mM or 10 mM 5-HT. Percentage survival across 21 day experiment. Significance determined via two-way ANOVA and Tukey's multiple comparisons test, *** $p \leq 0.005$, **** $p \leq 0.001$.
(DOCX)

**S7 Fig. (A)** Motility of 21 day old juvenile *Fasciola hepatica* after 24 hour exposure to 0.01 µM or 0.1 µM fluoxetine hydrochloride (FluHCl). (B) Motility of 28 day juvenile liver fluke maintained in 50% **chicken serum** (**CS**) supplemented with 1mM serotonin (5-HT). Each dot represents motility (length change mm/min) of individual juvenile fluke. Significance determined by Kruskal Wallis analysis and Dunn's multiple comparisons post hoc testing * p=≤0.05; ** p=≤0.01; **** p=≤0.001.
(DOCX)

**S8 Fig. Knockdown values (ΔΔCt) of serotonin signalling genes post RNA interference.** Values calculated from real time (q)PCR analysis using Pfaffl [92] equation and relative to glyceraldehyde 3-phosphate dehydrogenase (GAPDH) housekeeping gene.
(DOCX)

## Author contributions

**Conceptualization:** Emily Robb, Aaron G. Maule.

**Data curation:** Emily Robb, Sarah Muise, Lana Watt.

**Formal analysis:** Emily Robb, Sarah Muise, Lana Watt.

**Funding acquisition:** Nikki J. Marks, Aaron G. Maule.

**Investigation:** Emily Robb, Sarah Muise, Lana Watt.

**Methodology:** Emily Robb, Rebecca Armstrong, Duncan Wells, Paul McCusker.

**Project administration:** Emily Robb, John Harrington, Andreas Krasky, Paul M. Selzer, Nikki J. Marks, Aaron G. Maule.

**Resources:** Emily Robb, John Harrington, Andreas Krasky, Paul M. Selzer, Nikki J. Marks, Aaron G. Maule.

**Software:** Emily Robb.

**Supervision:** Nikki J. Marks, Aaron G. Maule.

**Validation:** Emily Robb.

**Visualization:** Emily Robb, Sarah Muise.

**Writing – original draft:** Emily Robb.

**Writing – review & editing:** Emily Robb, Sarah Muise, Lana Watt, Rebecca Armstrong, Duncan Wells, Paul McCusker, Nikki J. Marks, Aaron G. Maule.

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
