## [Decision Letter · Decision Letter 0]

11 Jan 2026

Dear Dr. Robb,

Thank you for submitting your manuscript to PLOS Neglected Tropical Diseases. After careful consideration, we feel that it has merit but does not fully meet PLOS Neglected Tropical Diseases's publication criteria as it currently stands. Therefore, we invite you to submit a revised version of the manuscript that addresses the points raised during the review process.

* A letter that responds to each point raised by the editor and reviewer(s). You should upload this letter as a separate file labeled 'Response to Reviewers '. This file does not need to include responses to any formatting updates and technical items listed in the 'Journal Requirements' section below.'. This file does not need to include responses to any formatting updates and technical items listed in the 'Journal Requirements' section below.

* A marked-up copy of your manuscript that highlights changes made to the original version. You should upload this as a separate file labeled 'Revised Manuscript with Track Changes '.'.

* An unmarked version of your revised paper without tracked changes. You should upload this as a separate file labeled 'Manuscript '.'.

We look forward to receiving your revised manuscript.

Kind regards,

Maria Y Pakharukova, Ph.D., D.Sc.

Academic Editor

Peter Fischer

Section Editor

Shaden Kamhawi

co-Editor-in-Chief

Paul Brindley

co-Editor-in-Chief

**Journal Requirements:**

2) Some material included in your submission may be copyrighted. According to PLOSu2019s copyright policy, authors who use figures or other material (e.g., graphics, clipart, maps) from another author or copyright holder must demonstrate or obtain permission to publish this material under the Creative Commons Attribution 4.0 International (CC BY 4.0) License used by PLOS journals. Please closely review the details of PLOSu2019s copyright requirements here: PLOS Licenses and Copyright. If you need to request permissions from a copyright holder, you may use PLOS's Copyright Content Permission form.

Potential Copyright Issues:

i) Please confirm (a) that you are the photographer of 6B, or (b) provide written permission from the photographer to publish the photo(s) under our CC BY 4.0 license.

ii) Figure 1C. Please confirm whether you drew the images / clip-art within the figure panels by hand. If you did not draw the images, please provide (a) a link to the source of the images or icons and their license / terms of use; or (b) written permission from the copyright holder to publish the images or icons under our CC BY 4.0 license. Alternatively, you may replace the images with open source alternatives. See these open source resources you may use to replace images / clip-art:

3) Kindly revise your competing statement in the online submission form to align with the journal's style guidelines: 'The authors declare that there are no competing interests.'

**Reviewers' comments:**

Reviewer's Responses to Questions

**Key Review Criteria Required for Acceptance?**

**Methods**

-Are the objectives of the study clearly articulated with a clear testable hypothesis stated?

-Is the study design appropriate to address the stated objectives?

-Is the population clearly described and appropriate for the hypothesis being tested?

-Is the sample size sufficient to ensure adequate power to address the hypothesis being tested?

-Were correct statistical analysis used to support conclusions?

-Are there concerns about ethical or regulatory requirements being met?

Reviewer #1: This study represents a large amount of well-executed work directed at characterization and manipulation of the 5-HT signaling pathway in Fasciola hepatica. Strengths of the work include the scope of the bioinformatic analysis, and RNAi/rescue experiments which bring important insight as to the endogenous role of different components of this signaling pathway. As such, this provides foundation for future expression, localization and RNAi assays to dissect the role of individual receptors in this system.

Reviewer #2: This manuscript provides new and original basic research into aspects of serotonin signalling in the trematode parsite, Fasciola hepatica. The authors provide a clearly laidout, rigorous body of work encompassing in silico analysis, molecular modelling, transcriptomics, along with growth and motility studies using juvenile flukes following 5-HT stimulation and gene silencing. This work is directly relevant to the search for new anthelmintics to treat F. hepatica infection in both humans and domestic animals. Although foundational, the data providec will serve as a platfom for further applied studies which have a reasonable likelihood of leading to potential new drug development strategies.

My answer is yes to all of the questions above, although perhaps more in depth statistical treatment / explanation might be warranted - were all of the responses indicative of a normal distribution, for example.

Reviewer #3: (No Response)

Reviewer #4: (No Response)

**Results**

-Does the analysis presented match the analysis plan?

-Are the results clearly and completely presented?

-Are the figures (Tables, Images) of sufficient quality for clarity?

Reviewer #1: Results are clearly presented throughout.

Reviewer #2: Yes

Results are clearly presented, although some of the figures are very detailed (network diagram with cluster analyssis) and consequently perhaps could be improved.

Reviewer #3: (No Response)

Reviewer #4: (No Response)

**Conclusions**

-Are the conclusions supported by the data presented?

-Are the limitations of analysis clearly described?

-Do the authors discuss how these data can be helpful to advance our understanding of the topic under study?

-Is public health relevance addressed?

Reviewer #1: Suggestions for clarification are provided in comments to authors.

Reviewer #2: Yes to all questions.

Reviewer #3: (No Response)

Reviewer #4: (No Response)

**Editorial and Data Presentation Modifications?**

Reviewer #1: None.

Reviewer #2: Suggestions on statistics and figure presentation are presented above - but these are minor in the context of the overall scope of the work.

Reviewer #3: (No Response)

Reviewer #4: (No Response)

**Summary and General Comments**

Reviewer #1: Major Comments

#1. Figure 5A, Worm motility analysis. The action of 5-HT on movement is shown after exposure to 1mM 5-HT for 1 day, a chronic exposure. Are stimulatory effects on motility seen under more acute conditions (mins to hours)? Presenting more acute times is important to show whether the phenotype represents an acute effect on neuromuscular signaling.

#2. The 5-HT stimulatory effect is evident through comparison to a basal medium lacking 5-HT. The authors should comment on the background 5-HT concentration in RPMI or RPMI/chicken serum, together with insight as to whether movement tracks higher in these media compared to the basal medium? This same thought applies to 5-HT concentration and growth effects (Figure 6) in different media.

#3. Figure 5. Interpretation of RNAi. The message seems to be that the 5HT(7) receptors are linked to motility (line 509) rather than ‘autoregulatory’ 5-HT(1) receptors. However, figure 5C shows statistically equivalent effects on motility after group 1 or group 7 RNAi. Figure 5E shows that basal motility is lower in the 5-HT(1) RNAi cohort rather than the 5-HT7 cohort, again suggesting that 5-HT(1) regulate movement. If 5-HT(1) was autoinhibitory for movement you would see a potentiation in basal movement in RNAi worms which is exactly the opposite than the effect reported on motility (Fig. 5E) as compared for growth (Fig. 6D). Some clarification here is needed.

#4. Functional characterization. The authors state (lines 263-265) that much of our understanding of 5HT GPCRs in parasitic flatworms has been driven by in silico approaches (citing a 20 year old review). I think we have a reasonable amount of data about parasitic flatworm 5-HT GPCRs - in fact they are probably the most studied flatworm GPCRs. For example, schistosome 5-HT GPCR has been subject to library screening (PMID: 30059006, PMID: 27187180 ), with insight into parasite versus human ligand specificities. Therefore, the statement that our knowledge base is only from in silico prediction should be better contemporized.

Minor Comments

#1 Introduction (line 97). In describing 5-HT GPCR coupling in vertebrates, the authors refer only to cAMP modulation. Human 5-HT GPCRs in addition couple to Gq/11, G12/13.

#2 Results and discussion. Lines 167/168. Use of ‘expression’ in context of mRNA is ambiguous, as ‘expression’ normally interpreted as mRNA translation that is as expressed proteins. Better to refer to mRNA abundance/levels.

#3 Line 181. Refer to location of these highly conserved motifs in terms of TM localization, or ideally Ballesteros-Weinstein numbering to help compare structures/function between different GPCRs. Referencing residue number in lines 291-298 also could be better anchored to allow future comparison.

#4 Line 253. GPCR number. I don’t really understand the argument being made here in terms of 5-HT GPCR number and chemosensation, unless the authors are proposing that these receptors are sensing 5-HT as a chemosensory cue. All presented evidence suggests that the 5-HT GPCRs are responding to parasite rather than environmental 5-HT. Please elaborate further or remove.

#5 Line 299-312. G protein coupling. Insert qualification that these are predictive couplings.

#6 Line 358. Seems to be missing a reference.

#7 Figure 5A, Worm motility analysis. The stimulatory effect of fluoxetine is suggested to result from inhibition of 5-HT reuptake. Fluoxetine has also been demonstrated to act as a flatworm 5-HT receptor blocker, so inhibition of a presynaptic autoreceptor could also explain potentiation.

#8 Line 488. Figure 5C?

#9 Line 488. 5HT-7 knockdown is stated to cause a 60% reduction in motility. This does not seem reflective of the data presented in Figure 5C where the reported mean is >50%.

#10 Some statement as to why 5-HT(7c) was selected for FISH as opposed to the other receptors/components would be helpful for readers.

Reviewer #2: In summary, a very well-presented manuscript, with results clearly presented and the discussion, even when data is not as expected, cogently argued.

Reviewer #3: The manuscript by Robb and colleagues describes the characterisation of the serotonin signalling pathway in Fasciola hepatica by in silico bioinformatic analysis and role serotonin plays in growth and motility using RNAi. The study includes a large amount of relevant data, derived from several tools. In short, the manuscript is extremely well written and warrants publication in PLoS NTD. I have only minor comments as below.

Line 268 & 303 – inconsistent text for Gi/o – subscript vs not

Line 358 – how does the immunocytochemistry compare between life stages?

Line 408 – what stage is being referred to here as the juveniles? The 3 week old flukes from figure 4?

Line 700 – how many parasites used for the different exposures? Were replicates carried out?

Line 712 – how many parasites per replicate?

Reviewer #4: This manuscript describes a study of serotonin (5-HT) signalling pathways in Fasciola hepatica. Firstly, general analyses (in silico, IHC and ISH) explored the overall expression of pathway components. Treatment and silencing experimental approaches were then employed to demonstrate a causal association between elements of the 5-HT pathway and parasite motility and growth. The authors conclude that the experimental findings illustrate a central role for serotonin signalling in both juvenile parasite motility and growth, suggesting potential for therapeutic exploitation for liver fluke control.

My comments are as follows:

The introduction is a well written review of the current state of knowledge around the role of the serotonin pathway in helminths. However, it is quite lengthy and I question the value of an entire paragraph dedicated to the association between 5-HT and metabolic activity, as this is not a focus of the experimental work in the manuscript?

Methods:

• How long are the metacercariae stored ‘until needed’ – this cannot be indefinite as viability will be affected – please note the maximum storage length that is used here.

• It is surprising that excysted Fasciola consistently survive for extended periods in the culturing environment described here (chicken sera and media). The authors should provide details (either here or in the supplementary section) on the numbers of excysted flukes that are cultured, the relative number that survive for 21 days and the range of sizes (growth rate) achieved for all fluke. Any interpretation of subsequent assessment of viability, motility, response to treatment or gene silencing, is totally dependent on this information, as any conclusion needs to be applicable to an entire population, not just the singular few that survive under these conditions. More information is required on the experimental set up and on the numbers/quality of parasites that are subsequently studied so it can be clearly understood how representative these flukes are.

• There is a distinct lack of transparency around the experimental design and analysis with no reference to numbers of worms/sections/slices examined throughout – either in the methodology or figure legends

Results/Discussion:

• The in silico analysis and modelling (Figures 1-3) follows a standard approach. However, it should be noted that the transcriptome data referred to in Figure 1 was performed on NEJs excysted in vitro (media only) and the juvenile flukes were 21 day old flukes isolated from host tissue. The authors have not determined whether the flukes they have maintained in culture for 21 days post excystment are closer in expression profile to the in vitro NEJs or the 21day ex vivo juveniles, or totally different. This is a critical basis for the interpretation of all subsequent data in terms of impact on treatment/accessibility/timing if proposing to target this pathway for parasite control.

• Figure 4 (legend and results text) – the authors refer to their cultured flukes as’ juveniles’ – have they confirmed this phenotype by comparison to flukes that have grown in a mammalian host? The use of this terminology needs to be accurate.

• Figure 5 – (A) why are newly excysted flukes examined here when the rest of the study focuses on flukes cultured for 21 days? (A-D) How many flukes per treatment group? How was motility determined – there is no description in either the methods of the results – was this quantification blinded?

• Figure 6 – the conclusion from this data is over-stated. While some parasites look to be larger in size, I would argue that this is only limited to a handful of the treated flukes – in general (photo and histogram) there is not a notable, or more importantly, a consistent increase in size. Once again, there is no clarity on the numbers of parasites assessed and on how they were selected for treatment/assessment making it difficult to properly interpret the results.

PLOS authors have the option to publish the peer review history of their article (what does this mean?If published, this will include your full peer review and any attached files.). If published, this will include your full peer review and any attached files.

**Do you want your identity to be public for this peer review?** For information about this choice, including consent withdrawal, please see our For information about this choice, including consent withdrawal, please see our Privacy Policy ..

Reviewer #1: No

Reviewer #2: No

Reviewer #3: No

Reviewer #4: No

**Figure resubmission:**
---

## [Decision Letter · Decision Letter 1]

26 Feb 2026

Dear Dr Robb,

We are pleased to inform you that your manuscript 'Serotonin signaling modulates growth and motility in juvenile Fasciola hepatica' has been provisionally accepted for publication in PLOS Neglected Tropical Diseases.

Best regards,

Maria Y Pakharukova, Ph.D., D.Sc.

Academic Editor

Peter Fischer

Section Editor

Shaden Kamhawi

co-Editor-in-Chief

Paul Brindley

co-Editor-in-Chief

Reviewer's Responses to Questions

**Key Review Criteria Required for Acceptance?**

**Methods**

-Are the objectives of the study clearly articulated with a clear testable hypothesis stated?

-Is the study design appropriate to address the stated objectives?

-Is the population clearly described and appropriate for the hypothesis being tested?

-Is the sample size sufficient to ensure adequate power to address the hypothesis being tested?

-Were correct statistical analysis used to support conclusions?

-Are there concerns about ethical or regulatory requirements being met?

Reviewer #1: (No Response)

Reviewer #3: (No Response)

Reviewer #4: (No Response)

**Results**

-Does the analysis presented match the analysis plan?

-Are the results clearly and completely presented?

-Are the figures (Tables, Images) of sufficient quality for clarity?

Reviewer #1: (No Response)

Reviewer #3: (No Response)

Reviewer #4: (No Response)

**Conclusions**

-Are the conclusions supported by the data presented?

-Are the limitations of analysis clearly described?

-Do the authors discuss how these data can be helpful to advance our understanding of the topic under study?

-Is public health relevance addressed?

Reviewer #1: (No Response)

Reviewer #3: (No Response)

Reviewer #4: (No Response)

**Editorial and Data Presentation Modifications?**

Reviewer #1: (No Response)

Reviewer #3: (No Response)

Reviewer #4: (No Response)

**Summary and General Comments**

Reviewer #1: Very thorough response to questions raised, with incorporated changes and additional clarifications very helpful. No remaining concerns.

Reviewer #3: The authors have addressed the comments raised by the review.

Reviewer #4: While several comments have been addressed the major outstanding issue is the presentation of worm size (Figure 5) as % area. It is not correct to normalise these data sets as it masks the variability between samples and experiments. The data should be presented as mm2; information which is currently within the supplementary data. It is unacceptable to manipulate the true outcomes of an experiment by normalising the readouts as is presented here.

PLOS authors have the option to publish the peer review history of their article (what does this mean?If published, this will include your full peer review and any attached files.). If published, this will include your full peer review and any attached files.

**Do you want your identity to be public for this peer review?** For information about this choice, including consent withdrawal, please see our For information about this choice, including consent withdrawal, please see our Privacy Policy ..

Reviewer #1: No

Reviewer #3: No

Reviewer #4: No

---

## [Editor Report · Acceptance letter]

Dear Dr Robb,

We are delighted to inform you that your manuscript, "Serotonin signaling modulates growth and motility in juvenile Fasciola hepatica," has been formally accepted for publication in PLOS Neglected Tropical Diseases.

Best regards,

Shaden Kamhawi

co-Editor-in-Chief

Paul Brindley

co-Editor-in-Chief
